# In vivo base editing rescues cone photoreceptors in a mouse model of early-onset inherited retinal degeneration

Elliot H. Choi[1,2,12], Susie Suh [1,2,12✉], Andrzej T. Foik [3], Henri Leinonen[1], Gregory A. Newby[4,5,6], Xin D. Gao[4,5,6], Samagya Banskota[4,5,6], Thanh Hoang[7], Samuel W. Du [1,8], Zhiqian Dong[1], Aditya Raguram[4,5,6], Sajeev Kohli[4,5,6], Seth Blackshaw[7], David C. Lyon[9], David R. Liu [4,5,6] & Krzysztof Palczewski [1,8,10,11✉]

Leber congenital amaurosis (LCA) is the most common cause of inherited retinal degeneration in children. LCA patients with *RPE65* mutations show accelerated cone photoreceptor dysfunction and death, resulting in early visual impairment. It is therefore crucial to develop a robust therapy that not only compensates for lost RPE65 function but also protects photoreceptors from further degeneration. Here, we show that in vivo correction of an *Rpe65* mutation by adenine base editor (ABE) prolongs the survival of cones in an LCA mouse model. In vitro screening of ABEs and sgRNAs enables the identification of a variant that enhances in vivo correction efficiency. Subretinal delivery of ABE and sgRNA corrects up to 40% of *Rpe65* transcripts, restores cone-mediated visual function, and preserves cones in LCA mice. Single-cell RNA-seq reveals upregulation of genes associated with cone photo-transduction and survival. Our findings demonstrate base editing as a potential gene therapy that confers long-lasting retinal protection.

[1] Gavin Herbert Eye Institute, Department of Ophthalmology, University of California, Irvine, CA, USA. [2] Department of Pharmacology, Case Western Reserve University, Cleveland, OH, USA. [3] International Centre for Translational Eye Research, Polish Academy of Sciences, Warsaw, Poland. [4] Merkin Institute of Transformative Technologies in Healthcare at Broad Institute, Cambridge, MA, USA. [5] Department of Chemistry and Chemical Biology, Harvard University, Cambridge, MA, USA. [6] Howard Hughes Medical Institute, Harvard University, Cambridge, MA, USA. [7] Department of Neuroscience, Johns Hopkins University School of Medicine, Baltimore, MD, USA. [8] Department of Physiology and Biophysics, University of California, Irvine, CA, USA. [9] Department of Anatomy and Neurobiology, School of Medicine, University of California, Irvine, CA, USA. [10] Department of Chemistry, University of California, Irvine, CA, USA. [11] Department of Molecular Biology and Biochemistry, University of California, Irvine, CA, USA. [12] These authors contributed equally: Elliot H. Choi, Susie Suh. ✉email: susie.suh@case.edu; kpalczew@uci.edu

Leber congenital amaurosis (LCA), the most common cause of inherited blindness in children, is characterized by progressive retinal degeneration and vision loss. LCA is caused by a loss-of-function mutation in one of several genes responsible for the development and function of the retina and/or the retinal pigment epithelium (RPE). To date, at least 15 different genes have been implicated in LCA, classifying LCA into different subtypes[1]. Of all subtypes, LCA type 2 (LCA2), caused by mutations in *RPE65*, accounts for up to 16% of all LCA cases[2]. The *RPE65* gene encodes a critical RPE-specific enzyme known as the retinal pigment epithelium-specific 65 kDa protein (RPE65), which converts all-*trans*-retinyl ester to 11-*cis*-retinol in a biochemical pathway called the visual cycle. Therefore, loss-of-function mutations in *RPE65* impair the visual cycle, contributing to retinal degeneration. A common clinical manifestation of LCA2 is early cone dysfunction and degeneration in the fovea and parafovea, which comprise the essential structure of the retina responsible for the highest visual acuity necessary for reading, driving, and performing many daily activities[3]. The mechanism of RPE65 deficiency leading to early cone degeneration is still unclear, but current evidence suggests that visual chromophore deficiency results in mislocalization of cone opsins, thereby contributing to cellular stress and susceptibility of the cones to degeneration[4]. Because the cone photoreceptors in the fovea are crucial for central vision and essential visual acuity, prevention of cone degeneration should be a primary goal for therapeutic intervention in LCA2 patients.

In recent years, gene therapy has taken major strides forward with successful clinical trials of *RPE65* augmentation therapy, which ultimately led to approval by the U.S. Food and Drug Administration[5–8]. The gene therapy delivers a full-sized functional copy of the *RPE65* transgene to patients' RPE cells using a serotype 2 adeno-associated virus (AAV) vector, and compensates for lost RPE65 function[9,10]. It has created new hope for LCA patients by improving their visual acuity. Yet, long-term outcomes regarding the progression of retinal degeneration are still controversial. Patients often showed a continuation of retinal degeneration and relapse in visual acuity a few years after treatment[11–14]. The cause of this progressive retinal degeneration is still unclear, but possible contributions include an inadequate or declining level of RPE65 expression from the delivered transgene or irreversible photoreceptor injury that cannot be reversed with RPE65 restoration[11–13]. These unmet needs highlight the importance of improving the current gene therapy approach, so that restored vision can be sustained for longer durations in patients by protecting photoreceptor cells from degeneration.

Previously by using a base editing strategy[15], we demonstrated restoration of visual function in a mouse model of LCA2 called *rd12* which has a nonsense T•A to C•G mutation in exon 3 of the *RPE65* gene. Base editors, which are comprised of the cytosine base editor (CBE) and the adenine base editor (ABE), enable direct conversion of a specific DNA base pair into another base pair (C•G to T•A or vice versa) at a targeted genomic locus, independent of Cas9-induced double-stranded DNA breaks[16,17]. Changes installed using base editors typically lead to precise outcomes, even in non-dividing cells, with minimal off-target effects. While we previously validated the rescue of visual function in the *rd12* mouse by base editing, we have not yet explored whether it can protect retinal photoreceptors from further deterioration as is observed in patients years after gene augmentation therapy. Here, we found that the base editing treatment can rescue the function and survival of cone photoreceptors on a long-term basis in the *rd12* mice, which exhibit rapid cone photoreceptor degeneration. Because protecting photoreceptors is key to preventing further deterioration of vision in LCA patients,

this study lays the foundation for establishing the therapeutic potential of base editing as a one-time, durable treatment for LCA2.

## Results

**Rd12 mice exhibit early cone dysfunction and rapid cone degeneration**. We first examined the time course of cone degeneration in the *rd12* mouse to determine the optimal age for treatment. In mice, cones are classified by two types of light-detecting opsins, S-opsin (short wavelength-sensitive or blue/ultraviolet (UV)-sensitive) and M-opsin (medium wavelength-sensitive or green-sensitive), which are expressed in an opposing dorsal–ventral gradient[18]. Previous studies have reported that mouse models of RPE65 deficiency display early cone dysfunction and degeneration from 2 weeks of age, even before complete development of the retina, and extensive loss of S-opsin-positive cones occurs by 5 weeks of age[19–21] (Fig. 1a).

At 3 weeks of age, the densities of S-opsin-positive cones (S-cones) and M-opsin-positive cones (M-cones) were already decreased, as shown on retinal flatmounts of *rd12* mice, in comparison to those of age-matched wild-type (WT) mice (Fig. 1b). Retinal cross-sections of *rd12* mice also revealed mislocalization of S-opsins and M-opsins to the cone inner segments, in contrast to the correct localization to the cone outer segments seen in WT mice (Fig. 1c, d). By 6 weeks of age, nearly all S-cones had disappeared from the retinal flatmounts (Fig. 1b, d), while M-cones still remained in the dorsal retina. The retinal cross-sections, however, revealed M-opsin mislocalization and shortening of the cone outer segments, indicating a progressive pathological process in the M-cones (Fig. 1c). Based on these findings, we decided to administer the treatment in mice at 3 weeks of age, by which time the retina is fully developed and the process of degeneration has already begun, and then evaluate the post-treatment outcome starting from 6 weeks of age.

**Evolved adenine base editor enhances the mutation correction rate in vitro**. Although we previously found that the adenine base editor (ABE) and single-guide RNA (sgRNA) can correct the *rd12* mutation, we now sought to improve the base editing efficiency by testing other ABE variants that can recognize a wider array of protospacer-adjacent motif (PAM) sequences. PAM sequences are short DNA sequences (2–6 base pairs) directly downstream of the DNA protospacer sequence targeted by the Cas9 nuclease in the CRISPR bacterial immune system. Therefore, having the correct PAM sequence at the target site is necessary for successful genome targeting. Previously, we reported that two sg RNAs, sgRNA-A5 (A5) and sgRNA-A6 (A6), which place the mutant base at the fifth and sixth base position of the protospacer, respectively, can correct the *rd12* mutation with codon-optimized ABE (7.10) coupled to the WT nSpCas9 (herein referred to as wtABE)[22], despite lacking the canonical NGG PAM sequence at the targeted site[15]. In the treated animals, A5 showed a higher on-target base editing efficiency, whereas A6 demonstrated a higher precision with lower bystander base editing. To enhance the on-target correction rate and reduce bystander base editing, we evaluated three other evolved ABE variants that were shown to be more compatible with the PAM sequences of A5 and A6; namely, codon-optimized NG-ABE, xABE, and NRRH-ABE[23,24] (Fig. 2a). We transfected different combinations of ABE and sgRNA into a cell line that stably expresses *Rpe65*[rd12] cDNA (*rd12* cell line) and then analyzed the base editing outcome by deep DNA sequencing.

The sequencing analysis revealed that regardless of the ABE type, co-transfection of A5 consistently showed a higher rate of bystander A-to-G conversion than co-transfection of A6,

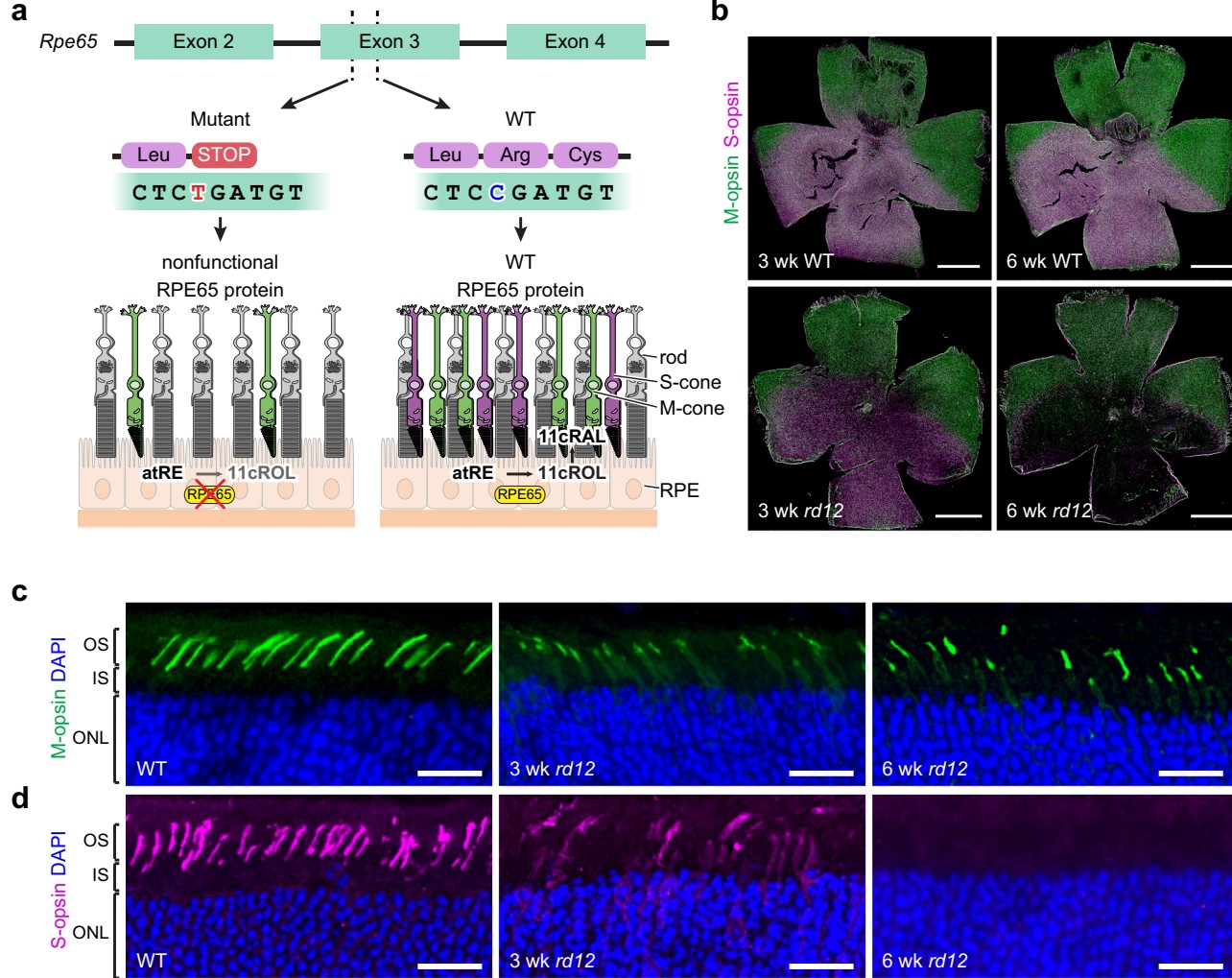

**Fig. 1 Early cone dysfunction and degeneration in the *rd12* mouse due to a loss-of-function mutation in *Rpe65*. a** The *rd12* mouse model has an inherent nonsense mutation (C•G to T•A) in exon 3 of the *Rpe65* gene, resulting in truncated, nonfunctional RPE65 protein. The deficiency of functional RPE65 in *rd12* mice impairs the production of 11-*cis*-retinal (11cRAL) by a blockade of conversion from all-*trans*-retinyl ester (atRE) into 11-*cis*-retinol (11cROL), and contributes to early cone cell death. **b** Retinal flatmounts from 3-week-old and 6-week-old wild-type (upper) and *rd12* (lower) mice, labeled with M-opsin (green) and S-opsin (purple) antibodies. The retina is oriented with the dorsal region towards the top and the ventral region towards the bottom. Scale bar, 1 mm. **c** Retinal cryosections representing the dorsal region from a 6-week-old wild-type (WT) mouse (left), 3-week-old *rd12* mouse (middle), and 6-week-old *rd12* (right) mouse, labeled with M-opsin (green) antibody. **d** Retinal cryosections representing the ventral region from the same eyes in **c**, labeled with S-opsin (purple) antibody. DAPI, blue. OS outer segment, IS inner segment, ONL outer nuclear layer. Scale bar, 20 μm.

especially at the adenine located 2 bases downstream of the target mutation (Fig. 2b). Among the transfected groups with A5, the wtABE showed the highest on-target base conversion ($12.76 \pm 0.03\%$), contrary to our expectation that NRRH-ABE would be more efficient at the recognition of A5 PAM (GAG). Among the transfected groups with A6, the NG-ABE showed the highest on-target correction rate ($27.64 \pm 0.20$), which was the highest of all the groups.

Because the true depth of rescue is determined by the relative amount of functional *Rpe65* alleles, we examined the percentage of precisely corrected *Rpe65* alleles, containing only the desired edit and no bystander missense mutations, in each transfection group (Fig. 2c). The A6 + NG-ABE group contained $24.36 \pm 0.26\%$ of functional *Rpe65* alleles, which was the highest percentage among all groups. The A6 + NG-ABE group contained $4.61 \pm 0.17\%$ of *Rpe65* alleles with bystander base edits. Given its superior correction rate and relatively low bystander editing, we selected the combination of A6 and NG-ABE to test in our animal models.

**Subretinal delivery of NG-ABE and sgRNA-A6 improves the correction rate**. To deliver the NG-ABE and sgRNA-A6 to the mouse RPE, we packaged these expression sequences into a single lentivirus vector (LV-NG-ABE-A6) (Fig. 2d), and injected the lentivirus subretinally into 3-week-old *rd12* mice. At 3-weeks-post-injection, we evaluated the outcome of base editing. Genomic DNA analysis from the treated RPE cells showed up to 57% of A-to-G conversion at the target adenine ($A_6$), with the average correction of $22 \pm 18\%$ ($n = 6$ eyes) (Supplementary Fig. 1). However, we noted that the process of isolating RPE cells from the posterior eyecup resulted in a variable distribution of RPE cells within each sample due to contamination with other cell types, hindering the DNA analysis exclusively within RPE cells. Therefore, we detached all cells in the posterior eyecup and examined the sequence of *Rpe65* cDNA, as *Rpe65* is exclusively expressed in the RPE cells.

The sequencing analysis of *Rpe65* cDNA from the treated eyes revealed up to 82% of A-to-G conversion at the target adenine ($A_6$) with an average frequency of $54 \pm 22\%$ ($n = 6$) (Fig. 2e). The

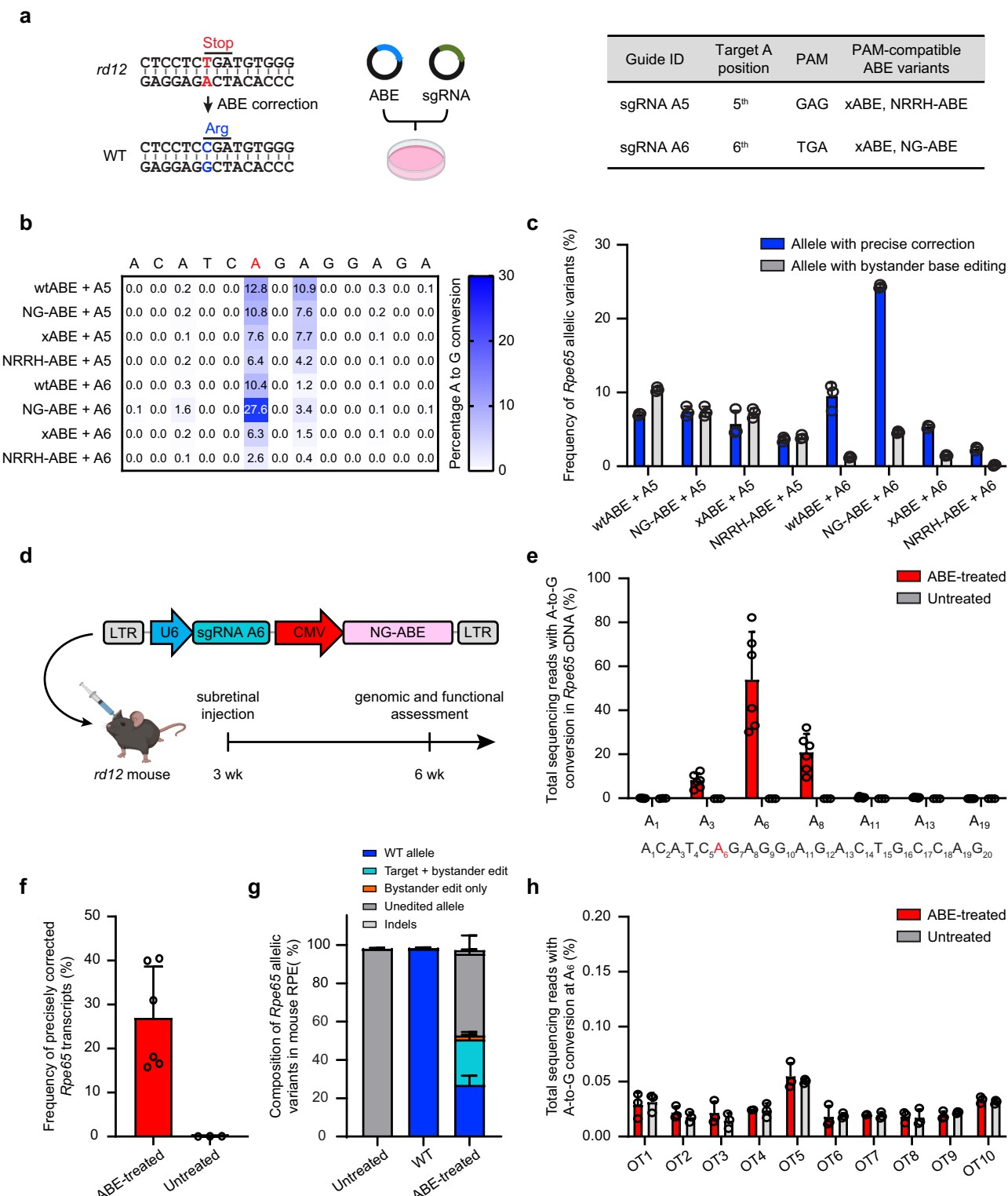

most frequent bystander editing occurred at $A_8$ ($21 \pm 8\%$), followed by $A_3$ ($8 \pm 3\%$), consistent with the pattern predicted from the in vitro study (Fig. 2b). When we examined the percentage of precisely corrected *Rpe65* transcripts within each treated eye lacking bystander edits, there was up to 40% of functionally rescued alleles in the eye, with an average frequency of $27 \pm 12\%$ in all eyes (Fig. 2f). Other modified transcripts were comprised of those containing A-to-G conversion of both mutation and bystander bases ($24 \pm 9\%$), or bystander bases only

($2 \pm 1\%$) (Fig. 2g). We also sequenced the top ten potential off-target sites identified by the CIRCLE-seq[25], but did not detect off-target editing above the background level of the untreated eyes (Fig. 2h and Supplementary Fig. 2). We further confirmed the expression of RPE65 protein via Western blot and the recovery of photoreceptor function by scotopic electroretinography (ERG) after dark adaptation (Supplementary Fig. 3). The treated mice showed scotopic a-wave and b-wave amplitudes of 68% and 74% of WT, respectively.

**Fig. 2 In vitro screening and in vivo validation of enhanced base editing with NG-ABE and sgRNA A6. a** Schematic representation of the in vitro strategy to screen different adenine base editors (ABEs) and sgRNAs capable of correcting the *rd12* mutation in the mutant cell line. The table on the right summarizes the choices of ABE and sgRNA for each transfection. **b** The heatmap shows the average frequency of A-to-G conversion in the genomic DNA isolated from the transfected cells. The *rd12* mutation is labeled in red. $n = 3$ biologically independent replicates, mean ± SD. **c** Percentage of *Rpe65* alleles that have a precise correction or contain bystander editing from each transfection group. $n = 3$ biologically independent replicates, mean ± SD. **d** Map of lentivirus (LV-NG-ABE-A6) for subretinal delivery of NG-ABE and sgRNA-A6 to the *rd12* mouse, and the experimental timeline. **e** Frequency of A-to-G conversion in the *Rpe65* cDNA isolated from lentivirus-injected (ABE-treated) and PBS-injected (untreated) *rd12* mouse eyes. The bottom sequence represents 20-nucleotide sgRNA-A6 with the targeted mutation highlighted in red. ABE-treated, $n = 6$; untreated, $n = 3$. Mean ± SD. **f** Frequency of precisely corrected *Rpe65* transcripts from the groups in **e**. **g** Average composition of *Rpe65* transcript variants in each eye from the untreated ($n = 3$), WT ($n = 3$), and ABE-treated ($n = 6$) mice. Mean ± SD. **h** Frequency of A-to-G conversion at ten potential off-target sites identified by CIRCLE-seq. $n = 3$ biologically independent replicates, each group. Mean ± SD. Source data are provided as a Source data file.

**AAV2-mediated delivery of NG-ABE/sgRNA A6 rescues the phenotype at a slower rate**. As AAV is the current vector of choice for gene therapy given its low immunogenicity and favorable safety profile, we also tested targeting the *rd12* mutation by packaging NG-ABE and sgRNA-A6 into AAV. Given the limited packaging capacity of AAV, we took advantage of a split base-editor dual-AAV strategy, in which the ABE is divided into amino-terminal and carboxy-terminal halves and packaged as two separate AAV-serotype 2 vectors[26] (Supplementary Fig. 4a). When both AAVs transduce the cell, protein splicing in trans reconstitutes a full-length base editor which can complex with the co-delivered sgRNA-A6 included in the AAV encoding the editor C-terminal fragment. We first examined the time course of AAV-mediated rescue by measuring scotopic ERG responses. The AAV-injected mice did not show detectable ERG responses until 7 weeks after injection (Supplementary Fig. 4b). Genomic DNA analysis of AAV-treated RPE showed relatively low base editing efficiency at the target mutation ($2.7 ± 1.2\%$), but the pattern of base editing was similar to that of lentivirus-treated RPE cells (Supplementary Fig. 4c). As well, the percentage of the precisely edited allele was the highest among all modified alleles with 1.6%, followed by alleles with both target and bystander editing at 0.9% (Supplementary Fig. 4d). Overall, our findings demonstrate that a split base-editor dual-AAV strategy can also correct the *rd12* mutation in mouse RPE cells, although it has a slower mode of action than lentivirus, which might be associated with AAV serotype, a requirement for two vectors, and an intracellular trans-splicing event. However, the dual AAV approach is not feasible for testing retinal deterioration in this mouse model due to the rapid degeneration of S-cones. Further optimization of the dual-AAV approach or new development of efficient and safe alternative delivery vehicles would pave the way for clinical translation of base editing technique to patients. Nevertheless, to test the ability of base editing for rescuing cone function and survival in mice, we opted to continue employing the lentiviral approach.

**Base editing restores cone-mediated visual function in the adult *rd12Gnat1*$^{-/-}$ mice**. The photoreceptors in the mouse retina are comprised of 98% rods and 2% cones[27]. Because a substantial contribution from rods makes it difficult to measure the visual function mediated only by cones, we eliminated the majority of rod-mediated photoresponse by crossing *rd12* mice to *Gnat1*$^{-/-}$ mice (*rd12Gnat1*$^{-/-}$), which lack the rod transducin α-subunit essential for the downstream signal transduction (Fig. 3a). In agreement with previous findings[28], the knockout of *Gnat1* did not have an impact on cone structure and survival, as shown on the retinal flatmount of the *Gnat1*$^{-/-}$ mouse (Supplementary Fig. 5). The *rd12Gnat1*$^{-/-}$ mouse showed a similar progression of cone degeneration to the *rd12* mouse, with S-cones decreasing from 2 weeks of age, and a complete degeneration by 6 weeks of age (Fig. 3b). We performed subretinal injection of

LV-NG-ABE-A6 into the *rd12Gnat1*$^{-/-}$ mice at 3 weeks of age and assessed cone function with photopic ERG using two distinct wavelengths of light at 6 weeks of age (Fig. 3c). The responses from M-cones were recorded by using a green light stimulus. Photopic ERG waveforms from eyes treated with LV-NG-ABE-A6 exhibited a prominent b-wave, which increased in amplitude with increasing stimulus intensities, whereas untreated eyes did not respond to any of the tested light intensities (Fig. 3d). Similarly, when we recorded the response from S-cones using a UV light stimulus, only the treated eyes showed ERG responses, which increased with higher intensities (Fig. 3e). The base editing treatment restored approximately 36% of M-cone function ($56.0 ± 11.0\,\mu V$) and 30% of S-cone function ($56.8 ± 11.6\,\mu V$) when compared with the age-matched *Gnat1*$^{-/-}$ eyes (M-cone, $157.5 ± 35.7\,\mu V$; S-cone, $187.5 ± 38.1\,\mu V$) (Supplementary Fig. 6).

In addition, we measured the functional integrity of the visual pathway from cones via the optic nerves to the visual cortex of the brain by recording visually evoked potentials (VEPs). The flash stimuli elicited distinct VEP waveforms consisting of three components (an initial negative deflection (N1), a positive deflection (P1), and a more variable negative deflection (N2)) in the control *Gnat1*$^{-/-}$ and treated *rd12Gnat1*$^{-/-}$ mice, but not in the untreated *rd12Gnat1*$^{-/-}$ mice (Fig. 4a, b). However, the VEPs in the treated mice showed attenuated amplitudes and delayed peak times, as compared to control mice (Fig. 4c, d). We hypothesize that the absence from the birth of visual chromophore in *rd12Gnat1*$^{-/-}$ mice could affect the normal development of visual cortex circuitry during the critical period. Therefore, intervention after the critical period is unable to completely restore normal visual cortical response in *rd12Gnat1*$^{-/-}$ mice. Nevertheless, the activities of single neurons in the primary visual cortex were also improved following the treatment (Fig. 4e, f).

**Base editing improves cone survival in the adult *rd12Gnat1*$^{-/-}$ mice**. To examine whether base editing prolongs cone survival in *rd12Gnat1*$^{-/-}$ mice, we measured the number of M-cones and S-cones on retinal flatmounts at 8 weeks of age, after staining with M-opsin- and S-opsin-specific antibodies. The overall view of retinal flatmounts showed remarkable preservation of S-cones in the treated eyes in comparison to the untreated eyes (Fig. 5a). Although M-cones seemed largely unchanged from an overall view of the retinal flatmounts between the treated and untreated eyes, a higher-magnification view of the mid-dorsal retina revealed both decreased density and structural abnormality of M-cones in the untreated retina compared to treated and control retinas (Fig. 5b). A higher-magnification view of the mid-ventral retina showed S-cones in the treated retina, whereas no S-cones were identified in the untreated retina (Fig. 5c). The survival of M-cones and S-cones was quantified by averaging the number of cones in five fields ($362 × 273\,\mu m$ each field) across the dorsal and ventral retina at 1 mm from the optic nerve in both treated

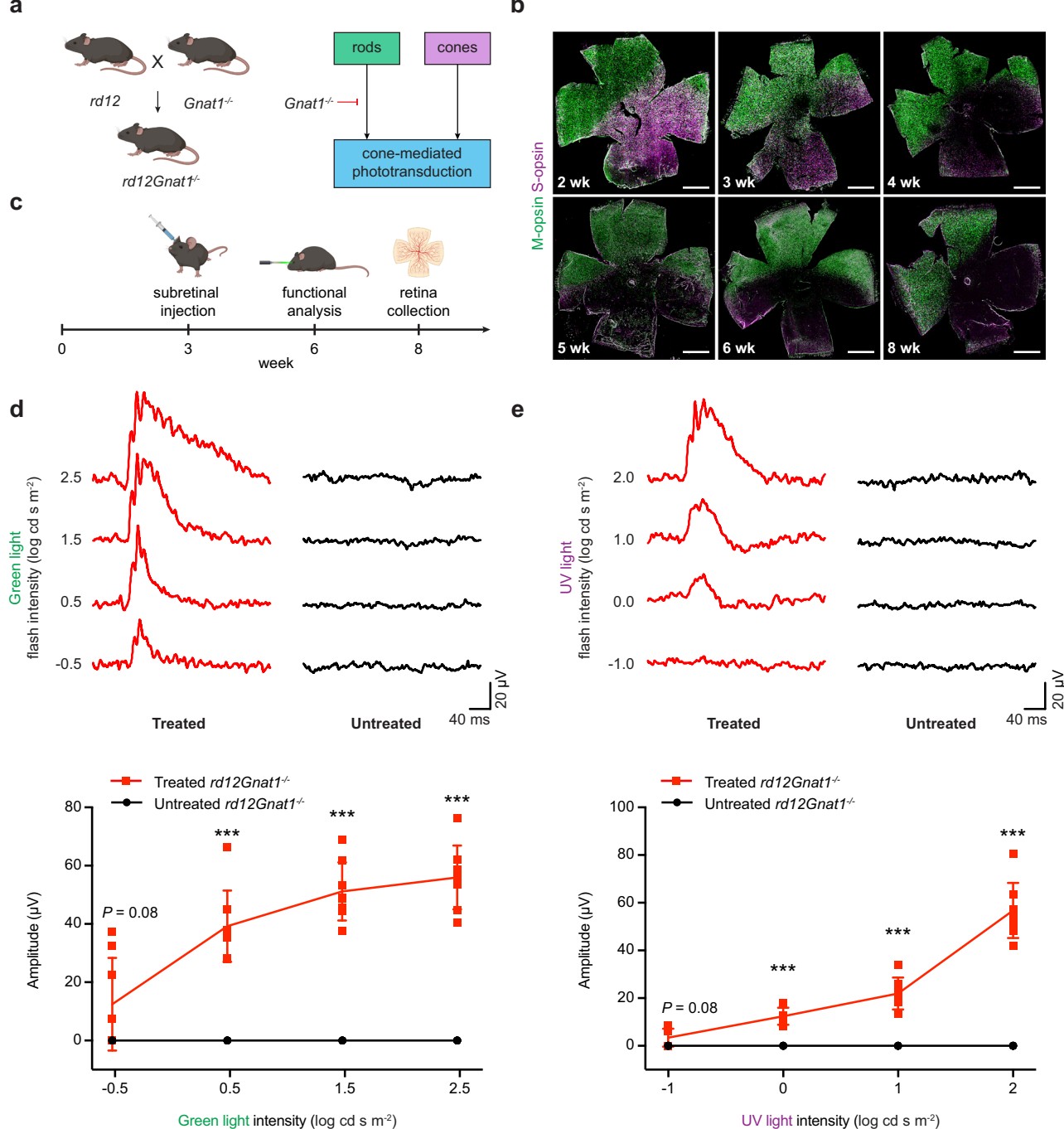

**Fig. 3 Rescue of cone-mediated visual function in *rd12Gnat1*<sup>−/−</sup> mice after ABE treatment. a** Breeding strategy to generate a homozygous *rd12Gnat1*<sup>−/−</sup> mouse line. Knockout (KO) of *Gnat1* nearly completely abolishes the phototransduction signaling cascade from rods, leaving cone-mediated phototransduction. **b** Progression of cone degeneration in the *rd12Gnat1*<sup>−/−</sup> mouse from 2 weeks of age to 8 weeks of age. M-opsin, green. S-opsin, purple. Scale bar, 1 mm. **c** Experimental timeline to evaluate cone function and survival in *rd12Gnat1*<sup>−/−</sup> mice after ABE treatment. **d** Representative photopic ERG waveforms of M-cones (top) and the average b-wave amplitudes (bottom), evoked with the indicated green light flashes in treated and untreated *rd12Gnat1*<sup>−/−</sup> mice (*n* = 8, each group). Mean ± SD. ***P < 0.001; two-tailed Mann–Whitney *U*-test. P values are 0.08 at −0.5 and 0.0002 at 0.5, 1.5, and 2.5 light intensities. **e** Representative photopic ERG waveforms of S-cones (top) and the average b-wave amplitudes (bottom), evoked with the indicated UV light flashes, in treated and untreated *rd12Gnat1*<sup>−/−</sup> mice (*n* = 8, each group). Mean ± SD. ***P < 0.001; two-tailed Mann–Whitney *U*-test. P values are 0.08 at −1 and 0.0002 at 0, 1 and 2 light intensities. Source data are provided as a Source data file.

and untreated eyes (*n* = 4 per group). In the treated eyes, the average number of S-cones per field was significantly higher in both dorsal (25 vs 5; *P* < 0.001) and ventral retina (123 vs 2; *P* < 0.001) compared to the untreated retina (Fig. 5d). The average number of M-cones per field was also significantly

higher in the dorsal (426 vs 194; *P* < 0.001) and ventral retina (61 vs 4; *P* = 0.006). Compared to the retinas of age-matched *Gnat1*<sup>−/−</sup> mice, retinas of the treated *rd12Gnat1*<sup>−/−</sup> mice showed protection up to 66% (vs. 30% in untreated) and 26% (vs. 2% in untreated) of M-opsins in the dorsal and ventral

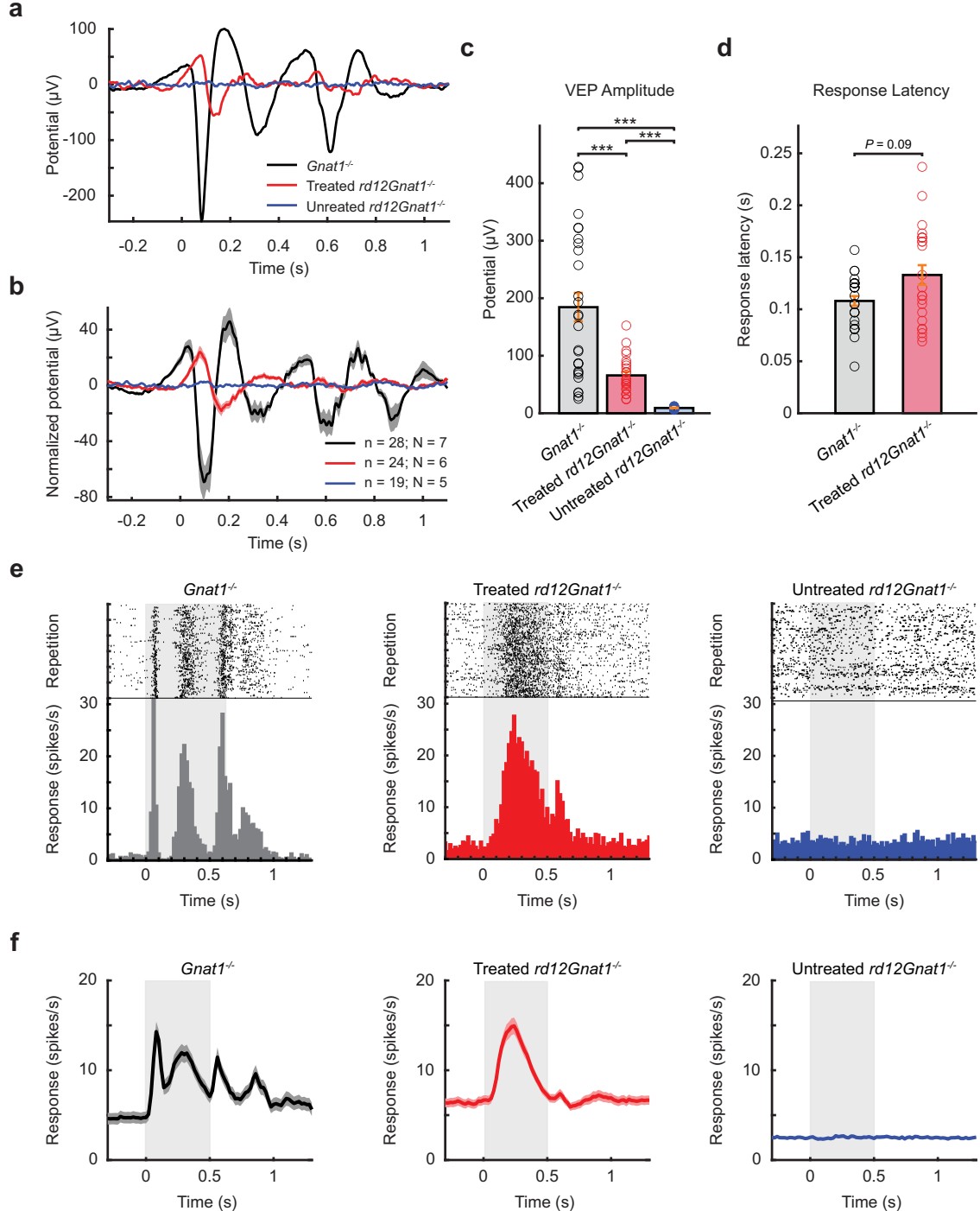

**Fig. 4 ABE treatment restores the visual pathway from cones to the visual cortex. a** Representative visually evoked potentials (VEPs) from control *Gnat1*$^{-/-}$, treated *rd12Gnat1*$^{-/-}$, and untreated *rd12Gnat1*$^{-/-}$ mice. **b** Population average of VEPs recorded from three groups in **a**. *N* total number of mice, *n* total number of recording sites from each experimental group. Shading indicates SEM. **c** VEP amplitudes recorded from three groups. Mean ± SEM. ***$P < 0.001$; two-tailed Mann–Whitney *U*-test. *P* values are 0.0004 for *Gnat1*$^{-/-}$ vs. treated *rd12Gnat1*$^{-/-}$ and <0.00001 for other comparison groups. **d** Response latencies of VEPs recorded from two groups. Mean ± SEM. Two-tailed Mann–Whitney *U*-test. **e** Raster plots and corresponding histograms for single neurons in *Gnat1*$^{-/-}$ (*n* = 387 cells), treated (*n* = 286 cells), and untreated *rd12Gnat1*$^{-/-}$ (*n* = 121 cells) mice, respectively. **f** Population average histograms for the same groups as **e**. Shading indicates SEM. Source data are provided as a Source data file.

region, respectively, and 25% (vs. 5% in untreated) and 17% (vs. 0.2% in untreated) of S-opsins in the dorsal and ventral region (Supplementary Fig. 6c, d). We also observed that both M-opsins and S-opsins were correctly localized to the cone outer segments in the treated *rd12Gnat1*$^{-/-}$ mice on retinal cryosections (Fig. 5e).

Long-term protection of cone function and structure was also examined in older mice at 6 months of age. ERG recordings from 6-month-old treated mice revealed clear preservation of light response, indicating continued M-cone and S-cone function (*n* = 4 per group) (Supplementary Fig. 7a, b). Furthermore, there was no significant decline in ERG amplitudes between 2 and

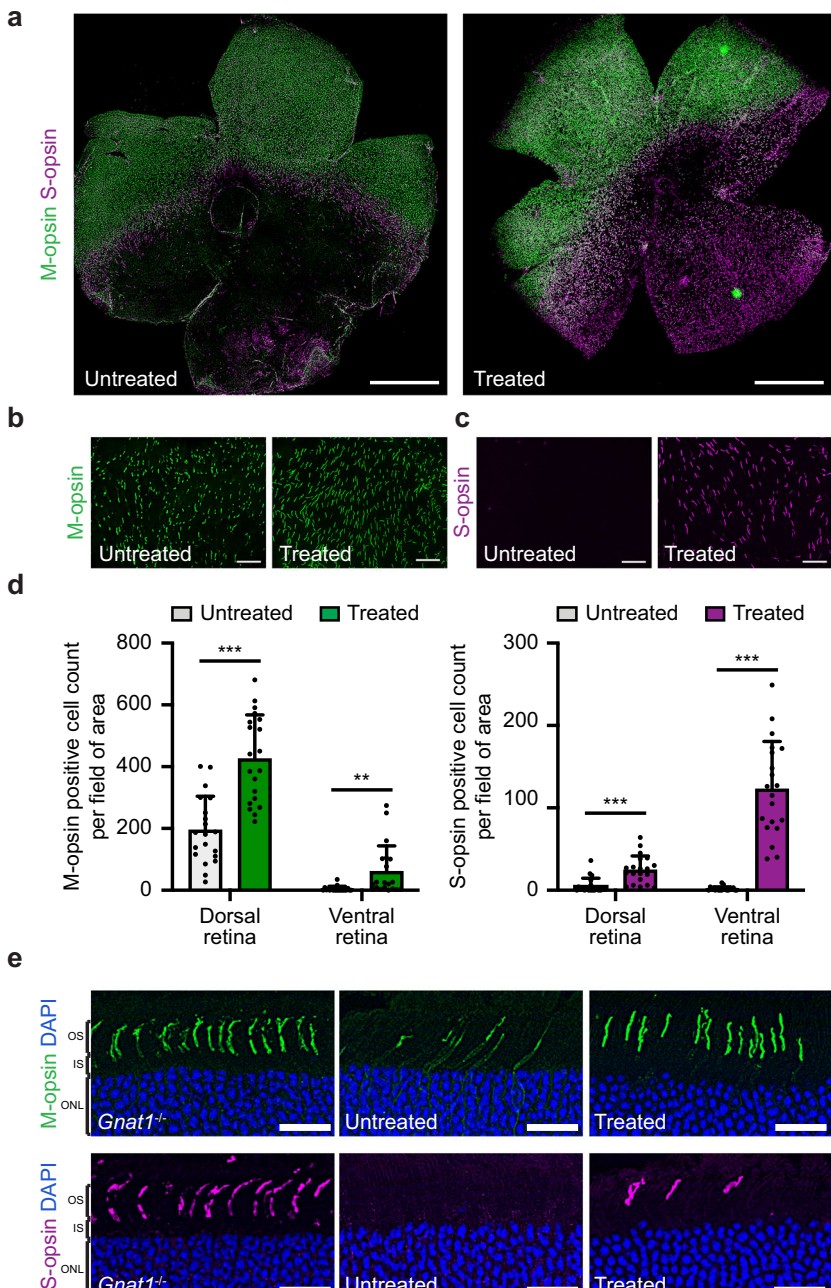

**Fig. 5 Protection of cone photoreceptors and correct opsin localization in 2-month-old *rd12Gnat1*⁻/⁻ following ABE treatment. a** Representative retinal flatmounts from untreated (left) and treated (right) *rd12Gnat1*⁻/⁻ mouse eyes, labeled with M-opsin (green) and S-opsin (purple) antibodies. Scale bar, 1 mm. **b** Magnified view of M-cones (green) in the dorsal retina from untreated and treated *rd12Gnat1*⁻/⁻ mice. Scale bar, 50 μm. **c** Magnified view of S-cones (purple) in the ventral retina from untreated and treated *rd12Gnat1*⁻/⁻ mice. Scale bar, 50 μm. **d** Quantification of M-cones and S-cones in each field as shown in **b**, **c**, at dorsal and ventral regions of the retina, 1 mm away from the optic nerve. Five fields across the dorsal or ventral retina were analyzed from each eye, with a total of 20 fields from 4 eyes per group. Mea ± SD. **P < 0.01; ***P < 0.001; unpaired *t*-test, two-tailed. *P* values for dorsal and ventral M-cones and dorsal and ventral S-cones are 0.000001, 0.006, 0.00009, and <0.000001, respectively. **e** Retinal cryosections from *Gnat1*⁻/⁻, and untreated and treated *rd12Gnat1*⁻/⁻ mice, showing the dorsal region labeled with M-opsin (top) and the ventral region labeled with S-opsin (bottom). DAPI, blue; OS outer segment, IS inner segment, ONL outer nuclear layer. Scale bar, 20 μm. Source data are provided as a Source data file.

6 months of age (Supplementary Fig. 7a, b). On retinal flatmounts from the treated eyes, both S-cones and M-cones were still detected at 6 months of age (Supplementary Fig. 7c, d). Cone quantification showed a significantly higher number of both M-cones and S-cones in the aged treated eyes, suggesting that base editing is able to prolong cone survival in the long-term (*n* = 3 per group) (Supplementary Fig. 7e).

**Base editing restores the expression of cone-specific photo-transduction genes and downregulates cell degeneration-associated genes.** To examine the impact of base editing on the transcriptional rescue of cone photoreceptors, we performed single-cell RNA-sequencing (scRNA-seq) of the retina with 2-month-old WT, untreated, and treated *rd12* mice (*n* = 4 retinas per group). We profiled 16,896 cells from three groups and

separated the cells into different clusters which were annotated by expression of cell type-specific marker genes[29] (Fig. 6a). Cell-type distribution was similar across the three groups (Fig. 6b). Cones formed a well-defined cluster, identified by the expression of the cone-specific marker (*Arr3*). At the transcriptome-wide level, both the cone and rod clusters of treated retinas more closely resembled the transcriptome profile of WT retinas than untreated retinas (Supplementary Fig. 8).

As expected, scRNA-seq showed a significantly increased expression level of *Opn1sw* (S-opsin) in the cone cells of the treated mice in contrast to the cone cells of the untreated mice ($P < 0.001$) (Fig. 6c and Supplementary Table 1). Interestingly, the expression level of *Opn1mw* (M-opsin) was higher in the retinas of both untreated and treated *rd12* mice compared to those of WT mice. We believe that this result is likely due to a greater fraction of captured cells being M-cones from the *rd12* mice as a result of early S-cone cell death that occurs even before treatment and which continues in unedited cells (Fig. 6c and Supplementary Table 1).

We found that expression levels of key genes involved in cone-specific visual phototransduction and bipolar cell synapses (*Arr3*, *Gnat2*, *Rbp3*, *Grk1*, and *Kcne2*) were notably downregulated in cone cells from untreated *rd12* mice compared to treated mice (Fig. 6d and Supplementary Table 1). In contrast, these genes all showed varying levels of rescued expression in the treated cone cells (Fig. 6d and Supplementary Table 1). In particular, cone arrestin (*Arr3*), (which along with *Opn1sw* showed the highest levels of rescue), is not only crucial for the regulation of the visual transduction cascade but is also essential for cone survival[30]. Loss of cone arrestin in a knockout mouse model was shown to increase the susceptibility of cones to cell death[31,32]. Therefore, we evaluated the expression of cone arrestin on retinal cryosections from treated and untreated *rd12* mice at 2 months of age. In untreated *rd12* mice, we could not detect cone arrestin even in the surviving cone cells, labeled with peanut agglutinin (PNA) (Fig. 6e). The treated *rd12* mice, on the other hand, showed expression levels of cone arrestin equivalent to age-matched WT mice (Fig. 6e), suggesting its implication on long-term cone protection.

In the context of increased cone cell-specific gene expression after treatment, we aimed to determine whether there was an associated change in gene transcripts related to cell survival, death, or stress. We would expect genes involved in promoting cone survival to be upregulated in treated samples relative to both WT and treated samples. Conversely, genes involved in cone cell death would be upregulated in untreated cones relative to both WT and treated cones. After three-way comparison and correction for false discovery, a total of 48 differentially expressed genes were identified (FDR < 0.05; Supplementary Table 2). Owing to the relatively small number of cones in the sample, the number of differentially expressed genes was not adequate to perform meaningful Gene Ontology or signaling pathway-enrichment analysis. However, a subset of these genes has been linked to either cell survival, cell death, or stress response. The only candidate neuroprotective gene in this group was *Mt1*, which was significantly upregulated in treated vs. untreated cones (Supplementary Fig. 9). *Mt2* was also strongly upregulated in cones of treated retinas but did not reach statistical significance due to its low abundance. *Mt1/2* are metal-chelating antioxidant metallothionein proteins that have been implicated in promoting photoreceptor survival and may prove to be useful biomarkers of treatment or potential therapeutic targets for further study[33].

Simultaneously, the treatment downregulated the expression of many genes that were significantly enriched in the untreated sample, and are therefore candidates for promoting photoreceptor cell death (Supplementary Fig. 9). These include the known pro-pyroptotic factor *Gsdme*[34]. The role of other genes in this category is less clear, with several consisting of regulators of cell adhesion (*Itga4*, *Edil3*), intracellular trafficking (*Rebepk*, *Chmp4b*), or isoprenoid biosynthesis (*Pmvk*). Surprisingly, this also includes *Fgf2*, which, though known to be induced strongly by photoreceptor injury[35], has generally been thought to be neuroprotective.

Although the treatment led to upregulation of genes linked to cone function and survival and downregulation of genes linked to cell death and stress, it is yet unclear whether altered gene expression is a contributing factor to or a subsequent manifestation of cone degeneration. This distinction would require further investigation with a larger sample group and more targeted analysis. Nevertheless, the findings from scRNA-seq demonstrate that base editing can reverse the altered gene expression profile of dysfunctional cones, further suggesting its long-term therapeutic benefit for photoreceptor protection.

## Discussion

Over the past several years, base editing has rapidly emerged as a potential approach to treat genetic disorders, with promising outcomes in different preclinical models[15,36–40]. The base editing approach has great promise especially for targeting genetic eye disorders, given the unique therapeutic advantages of the eye (immune privilege, accessibility, anatomical structure), and the prior demonstration of successful genetic rescue in a mouse model[15]. Previously, we demonstrated that subretinal delivery of ABE can correct the LCA mutation in a mouse model, suggesting its potential as a treatment. Here, we sought to answer whether base editing could rescue cone photoreceptor's survival and function, using an LCA mouse model. Prevention of further retinal degeneration in LCA patients has been a long-standing challenge, and therefore addressing this concern is highly important in the development of new therapeutic strategies.

To test our hypothesis, we selected the *rd12* mouse model, which displays an early and rapid degeneration of cone photo-receptors and mislocalization of cone opsins due to RPE65 deficiency. Because cone death occurs dramatically and rapidly in *rd12* mice, it served as an ideal model to evaluate the effectiveness of base editing for cone protection. In vitro transfection allowed us to predict the outcome of in vivo base editing mediated by different pairs of ABEs and sgRNAs combinations, and to identify the most efficient pair. This approach revealed the importance of screening multiple ABE variants and sgRNAs for each target sequence, as a single base difference in sgRNA can have a profound impact on the extent of rescue, and the most active ABE variant cannot always be predicted a priori. To evaluate the treatment effects on cone function and survival of mice, we delivered ABE and sgRNA via subretinal injections into $rd12Gnat1^{-/-}$ mice, which lack rod-transducin, rendering these mice cone function-dominant. Following treatment, we observed a substantial rescue of ERG activity in response to green and UV monochromatic light stimuli under light-adapted conditions. However, a caveat in using the *Gnat1* knockout strategy pertains to the finding that *Gnat2* (cone transducin) may support some residual rod-function[41], which raises the possibility that our treatment effect on ERG is partially mediated by rods. However, both M- and S-cones were anatomically preserved in treated mice up to 6 months of age, and single-cell RNA-seq analysis of treated retinas revealed the restoration of expression of genes associated with cone phototransduction and cone survival. Collectively, these results support the conclusion that the base editing strategy prolongs cone survival, transforms the gene expression signature of early-onset cone degeneration, and supports cone-mediated phototransduction.

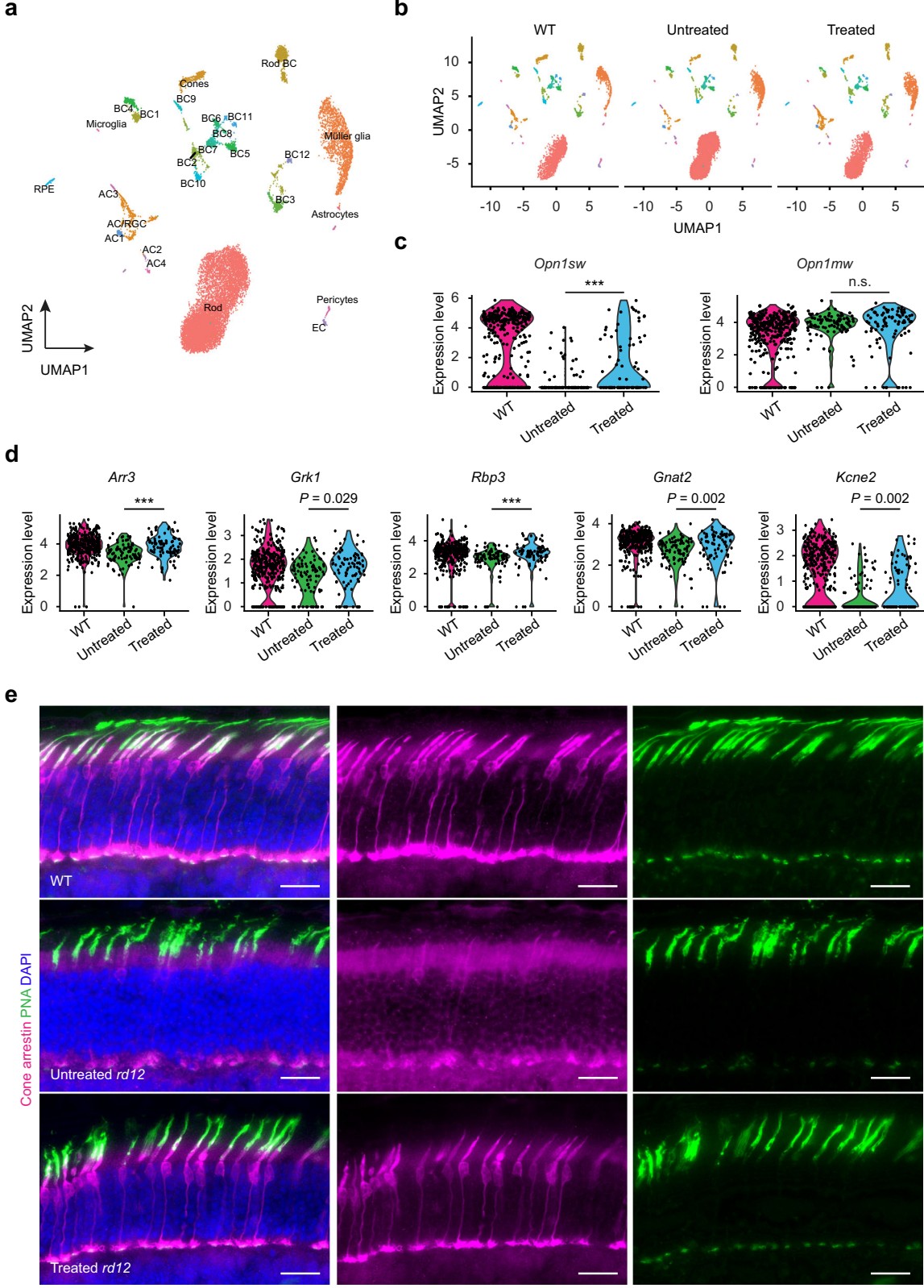

**Fig. 6 Single-cell RNA-seq of ABE-treated retinas reveals the rescue of genes associated with phototransduction and cone survival. a** Uniform manifold approximation and projection (UMAP) of the single-cell RNA-seq dataset colored by annotated cell type. **b** UMAP plots showing the distribution of single cells of 2-month-old WT, and untreated and treated *rd12* retina samples (*n* = 4, each group). **c** Violin plot profiles of S-opsin (*Opn1sw*) and M-opsin (*Opn1mw*) expression levels in individual cone cells. **d** Violin plot profiles of cone phototransduction-associated expression levels in individual cone cells. ***\*\*\*P* < 0.001; two-tailed Wilcoxon rank-sum test in Seurat. *P* values for **c**, **d** are provided in Supplementary Table 1. **e** Retinal cryosections showing the expression of cone-arrestin (purple) in 2-month-old WT, and untreated and treated *rd12* mice. Cone cells are labeled with peanut agglutinin (PNA; green). DAPI, blue. Scale bar, 20 μm.

Several factors can be identified as likely contributors to the robust rescue of the cone photoreceptors by base editing. First, base editing introduces a permanent genomic edit, thereby eliminating the possibility of diminishing expression from an episomal transgene over time. Secondly, it generates a more physiological pattern of gene expression, as the corrected gene is controlled by the endogenous promoter and chromatin context. As well, the survival of rod photoreceptors may lead to increased support for cone photoreceptor survival. Lastly, it eliminates the expression of the disease-causing truncated, dysfunctional protein, reducing cellular stress. Taken together, we believe that these factors were responsible for the sustained rescue of cone photoreceptors seen in the LCA2 mouse model.

In conclusion, we have shown significant protection against cone loss in a mouse model of LCA2 by base editing. Our results indicate that base editing is an effective therapeutic intervention that could be developed to rescue and sustain cone photoreceptors in humans in the context of inherited retinal degeneration. However, additional preclinical testing in non-human primates, which have foveas, would be necessary before this approach can be tested in patients. Furthermore, alternative delivery methods should be investigated to cover a broader area of the RPE tissue in patients and to circumvent constitutive base editor expression, which could induce unwanted DNA/RNA editing and immune reactions in the long term. With these further developments, base editing will provide a new paradigm for the treatment of numerous inherited ocular diseases caused by different modes of inheritance.

## Methods

**Ethics statement**. The Institutional Biosafety Committee (IBC) at the University of California, Irvine (UCI), has reviewed and approved the cell line experiments described in this study. The study was also conducted in accordance with NIH Guidelines and UCI Policy.

**Mice**. The *rd12* mice and C57BL/6J mice were purchased from the Jackson Laboratory (Jackson Laboratory; Bar Harbor; 005379 and 000664, respectively). *Gnat1*−/− mice were a generous gift from Janet Lem (Tufts University, Boston). *Rd12Gnat1*−/− mice were generated by crossbreeding *Gnat1*−/− mice with *rd12* mice. Progeny was genotyped as described previously[42]. The homozygosity of the *rd12* mutation was validated by Transnetyx genotyping. The age, strain, and the number of animals used for each experiment are stated in the corresponding figure captions. The ratio of males to females was relatively equal in all experiments. All mice were housed in the vivarium at the University of California, Irvine, where they were maintained on a normal mouse chow diet and a 12 h/12 h light (<10 lux)/dark cycle. The temperature ranged from 75 to 76 °F, and the humidity from 30 to 40%. All animal procedures were approved by the Institutional Animal Care and Use Committee (IACUC) of the University of California, Irvine, under Protocol #AUP-21-096 and were conducted in accordance with the Association for Research in Vision and Ophthalmology Statement for the Use of Animals in Ophthalmic and Visual Research.

**Cell line generation**. A stable cell line expressing a mouse RPE65[rd12] variant was generated by transduction of NIH3T3 cells with retrovirus obtained from Phoenix-Eco cells transfected with pMXs-RPE65(*rd12*)-IRES-GFP according to a previously published protocol[43].

**In vitro base editing validation**. NIH3T3-RPE65 (*rd12*) cells were seeded on a 24-well plate 18 h prior to transfection. At ~70% confluency, cells were transfected with 750 ng of ABE-expression plasmid and 250 ng of sgRNA-expression plasmid using 1.5 µl of Lipofectamine 3000 (Thermo Fisher, no. L3000001) per well. Four kinds of ABE-expression plasmids were used: pCMV-ABEmax (Addgene plasmid #112095), NG-ABEmax (Addgene plasmid #124163), xABEmax (Addgene plasmid #119813) and pCMV-ABEmax-NRRH[23]. Two sgRNA-expression plasmids were generated as previously described[15]. Cells were harvested for genomic DNA purification 48 h post-transfection.

**Lentivirus generation for in vivo ABE delivery**. To generate a single lentiviral vector co-expressing sgRNA-A6 and NG-ABEmax, the lentiviral transfer plasmid, LV-ABEmax-A6, generated during a previous study[15], was double-digested with EcoRI and Eco32I to replace a 2,284-bp sequence with the homologous sequence from NG-ABEmax (Addgene plasmid #124163), double-digested with EcoRI and

Eco32I. The final cloned plasmid was packaged into lentivirus particles from Signagen.

**AAV generation for in vivo ABE delivery**. N-terminal ABE7.10 AAV is identical to that published in a previous study[26]. To replace the C-terminal AAV plasmid Cas9 variant with SpCas9-NG, first, SpCas9-NG was amplified from NG-ABEmax (Addgene plasmid #124163) with the following primers: forward: TGCTTCGAC TCCGTGGAAATCTC and reverse: GACTTTCCTCTTCTTCTTGGGC; and the resulting product was cloned via Gibson assembly into C-terminal ABE7.10 AAV[26] that was double-digested with PasI and EcoRI. After sequence confirmation, the plasmid was digested overnight with BsmBI to insert the guide sequence. The guide sequence was ordered as two oligos which were annealed and phosphorylated in vitro before ligation into the cut vector using T4 DNA ligase. The sequences of the two oligos encoding the guide sequence were CACCGACATCAGAGGAGA CTGCCAG and AAACCTGGCAGTCTCCTCTGATGTC. Adeno associated viruses expressing the split base editor were produced using the previously described protocol[26].

Briefly, HEK293T/17 cells were plated in 15 cm dishes to about 80-85% confluency 24 h before transfection. Cells were then transfected with PEI containing 5.7 µg AAV genome, 11.4 µg pHelper (Clontech), and 22.8 µg of rep-cap plasmid per 15 cm dish. The medium was changed to DMEM with 5% fetal bovine serum 1 day after transfection. The virus was then extracted from cells 72 h after transfection from both the cell lysate and the supernatant. All the viruses were purified with an iodixanol step gradient using a Ti 70 fixed angle rotor at 58,600 rpm for 2 h, 15 min at 4 °C. Ultracentrifugation was followed with buffer exchange and a concentration step using 100-kD MWCO columns (EMD Millipore). The concentrated viral solution was sterile-filtered using a 0.22 µm filter, and stored at −80 °C until use. All viruses were titered via quantitative PCR using the AAVpro Titration Kit v.2 (Clontech), following the manufacturer's protocol.

**Targeted deep sequencing analysis**. Genomic DNA (gDNA) from cultured cells or mouse RPE tissue was isolated using the DNeasy Blood and Tissue Kit (Qiagen, no. 69504) or AllPrep DNA/RNA Mini Kit (Qiagen, no. 80284). Total RNA was extracted from the mouse posterior eye cup using AllPrep DNA/RNA Mini Kit (Qiagen, no. 80284). Complementary DNA (cDNA) was synthesized from the total RNA using Superscript III first-strand synthesis SuperMix (Thermo Fisher Scientific, no. 18080400). From gDNA and cDNA templates, 200–300 bp PCR amplicons of on- and off-target predicted sites for *Rpe65* were generated using primers with partial Illumina adapter sequences and then purified using the QIAquick PCR Purification Kit (Qiagen, no. 28106). Samples were sequenced on an Illumina MiSeq. Between 70,000 and 100,000 NGS reads for each sample were generated on single-end (1 × 150 bp) or paired-end (2 × 250 bp) runs. Primers used for off-target analysis are listed in Table S2.

**CIRCLE-seq off-target editing analysis**. Genomic DNA from *rd12* mouse tissue was isolated using the Gentra Puregene Tissue Kit (Qiagen, no. 158667), according to the manufacturer's protocol. CIRCLE-seq was performed as previously described[25,44]. Briefly, purified genomic DNA was sheared with a Covaris S2 instrument to an average length of 300 bp. The fragmented DNA was end-repaired, A-tailed, and ligated to a uracil-containing stem-loop adaptor, using KAPA HTP Library Preparation Kit, PCR Free (KAPA Biosystems). Adaptor-ligated DNA was treated with Lambda Exonuclease (NEB) and *E. coli* Exonuclease I (NEB) and then with USER enzyme (NEB) and T4 polynucleotide kinase (NEB). Intramolecular circularization of the DNA was performed with T4 DNA ligase (NEB) and residual linear DNA was degraded by Plasmid-Safe ATP-dependent DNase (Lucigen). In vitro cleavage reactions were performed with 250 ng of Plasmid-Safe-treated circularized DNA, 90 nM of Cas9-NG protein, Cas9 nuclease buffer (NEB), and 90 nM of synthetic chemically modified sgRNA (BioSpring) in a 100 µl volume. Cleaved products were A-tailed, ligated with a hairpin adaptor (NEB), treated with USER enzyme (NEB), and amplified by PCR with barcoded universal primers NEBNext Multiplex Oligos for Illumina (NEB), using Kapa HiFi Polymerase (KAPA Biosystems). Libraries were sequenced with 150-bp paired-end reads on an Illumina MiSeq. CIRCLE-seq data analyses were performed using open-source CIRCLE-seq analysis software (https://github.com/tsailabSJ/circleseq), using the following parameters: read_threshold: 4; window_size: 3; mapq_threshold: 50; start_threshold:3; gap_threshold: 3; mismatch_threshold: 6; search_radius: 30; PAM: NG; merged_analysis: True. The mouse genome GRCm38 was used for alignment.

**Mouse subretinal injection**. Mice were anesthetized by intraperitoneal injection of a cocktail consisting of 20 mg/ml ketamine and 1.75 mg/ml xylazine in phosphate-buffered saline at a dose of 0.1–0.13 ml per 25 g body weight, and their pupils were dilated with topical administration of 1% tropicamide ophthalmic solution (Akorn, no. 17478-102-12). Subretinal injections were performed using an ophthalmic surgical microscope (Zeiss). An incision was made through the cornea adjacent to the limbus at the nasal side using a 26-gauge needle. A 34-gauge blunt-end needle (World Precision Instruments, no. NF35BL-2) connected to an RPE-KIT (World Precision Instruments, no. RPE-KIT) by SilFlex tubing (World Precision

Instruments, no. SILFLEX-2) was inserted through the corneal incision while avoiding the lens and advanced through the retina. Each mouse was injected with 1 μl of viral vector prep per eye. Only mice that had more than 95% retinal coverage after subretinal injection and with minimal complications were kept for further evaluation.

**Western blot analysis**. To prepare the protein lysate from the mouse RPE tissue, the dissected mouse eyecup, consisting of RPE, choroid, and sclera, was transferred to a microcentrifuge tube containing 30 μl of RIPA buffer with protease inhibitors, homogenized with a motorized tissue grinder (Fisher Scientific, no. K749540-0000) and centrifuged for 30 min at $20,000 \times g$ at 4 °C. The resulting supernatant was pre-cleared with Dynabeads Protein G (Thermo Fisher, no. 10003D) to remove contaminants from blood. Twenty microliters of *rd12* cell lysates (15 μl for RPE lysates) were mixed with NuPAGE LDS Sample Buffer (Thermo Fisher, no. NP0007) and NuPAGE Sample Reducing Agent (Thermo Fisher, no. NP0004), incubated at 70 °C for 10 min, separated using a NuPAGE 4–12% Bis-Tris gel (Thermo Fisher, no. NP0321BOX), transferred onto a PVDF membrane (Millipore, no. IPVH00010), followed by blocking for 1 h in 5% (w/v) non-fat milk in PBS containing 0.1% (v/v) Tween-20 (PBS-T). The membrane was incubated with primary antibody diluted in 1% (w/v) non-fat milk in PBS-T overnight at 4 °C. Primary antibodies included mouse anti-RPE65 monoclonal antibody (1:1000; in-house production); mouse anti-Cas9 monoclonal antibody (1:1000; Invitrogen, no. MA523519, clone 7A9); and rabbit anti-β-actin polyclonal antibody (1:1000; Cell Signaling Technology, no. 4970S). After overnight incubation, membranes were washed three times with PBS-T for 5 min each and then incubated with secondary antibody for 1 h at room temperature. Secondary antibodies included goat anti-mouse IgG-HRP antibody (1:5000; Cell Signaling Technology, no. 7076S) and goat anti-rabbit IgG-HRP antibody (1:5000; Cell Signaling Technology, no. 7074S). After washing the membrane three times with PBS-T for 5 min each, protein bands were visualized using Odyssey XF Imaging System (Li-Cor) after exposure to SuperSignal West Pico Chemiluminescent substrate (Thermo Fisher, no. 34580).

**Immunohistochemistry of retinal flatmounts and cone quantification**. Mouse eyes were fixed with 4% paraformaldehyde in PBS (Santa Cruz Biotechnology, no. 30525-89-4) for 1 h at room temperature and washed three times in PBS for 10 min each. To make retinal flatmounts, the retinal tissue was separated from the anterior segment and posterior eyecup under a dissecting microscope, and four radial cuts were made toward the optic nerve head to flatten the retina. Retinal flatmounts were washed in wash buffer (PBS containing 0.5% Triton X-100 (Sigma-Aldrich, no. X100-5ML) three times for 5 min. To stain cone photoreceptors, retinal flatmounts were incubated with 5% normal donkey serum (Millipore Sigma, no. S30-100ML), polyclonal goat anti-S-opsin (1:500; custom-made by Bethyl Laboratories), and polyclonal rabbit anti-M-opsin (1:500; Novus Biologicals, no. NB110-74730) antibodies in wash buffer for 3 nights at 4 °C. Samples were washed three times for 5 min each in wash buffer and incubated with secondary antibodies, Alexa Fluor 488-conjugated donkey anti-goat IgG (1:250; Abcam, no. ab150129) and Alexa Fluor 647-conjugated donkey anti-rabbit IgG (1:250; Abcam, no. ab150075), for 2 h at room temperature in the dark. After final washing, samples were mounted on slides with VECTASHIELD antifade mounting medium (Novus, no. H-1000-NB). To count the number of S-cones and M-cones in a retinal flatmount, 5 images each of the dorsal and ventral retina were taken, approximately 1 mm away from the optic nerve using a ×40 objective lens in a Keyence BZ-X810 All-in-One fluorescence microscope. Each field ($362 \times 273$ μm) was captured with GFP and Cy5 filters to distinguish S-opsin and M-opsin, respectively. Automated cone quantification in each field was performed using the ImageJ software. All images were converted to RGB stack, and the size and intensity threshold was set to identify cone-opsin-positive cells.

**Immunohistochemistry of retinal cryosections**. Following enucleation, the cornea and lens were carefully removed under a dissecting microscope while maintaining the shape of the eyecup. The eyecup was fixed with 4% paraformaldehyde in PBS (Santa Cruz Biotechnology, no. 30525-89-4) for 2 h and washed with 5% sucrose in PBS three times for 5 min. The eyecup was dehydrated with 20% sucrose in PBS, embedded in 20% sucrose in O.C.T. (1:2 volume ratio, Sakura, no. 4583), and then flash-frozen for cryosectioning at 10 μm thickness. For immunostaining, cryosections were first blocked with 5% normal donkey serum in 0.2% Triton X-100 in PBS, and then incubated with primary antibodies overnight at 4 °C. The S-opsin and M-opsin antibodies were identical to those used for retinal flatmount staining. Cone arrestin was probed with polyclonal rabbit anti-cone arrestin antibody (1:400; Millipore Sigma, no. AB15282). Cone sheaths were stained with fluorescein-conjugated peanut agglutinin (1:200; Vector Laboratories, no. FL-1071). Secondary antibodies were identical to those used for retinal flatmount staining. After incubation for 2 h at room temperature with secondary antibodies, cryosections were washed three times for 5 min each before mounting with a mounting medium containing DAPI (Vector Laboratories, no. H-1500-10) and securing with a coverslip.

**Electroretinography**. Scotopic ERG recording was performed as previously described[15]. Before photopic ERG recordings, mice were kept in a lighted vivarium.

After induction of anesthesia, pupils were dilated with 1% tropicamide (Akorn, no. 17478-102-12), and bathed with 2.5% hypromellose (Akorn, no. 9050-1) to keep corneas hydrated. A mouse was placed on a heated Diagnosys Celeris rodent ERG device (Diagnosys LCC), and the ocular electrodes and ground electrode were placed on the corneas and hind leg, respectively. To measure M-cone and S-cone function, stimulation was performed with alternating green light and UV light at increasing intensities. Green light stimulation (peak emission 544 nm, bandwidth 160 nm) had intensity increments of −0.5, 0.5, 1.5, and 2.5 log cd·s/m². UV light stimulation (peak emission 370 nm, bandwidth 50 nm) had intensity increments of −1, 0, 1, and 2 log cd·s/m². The responses for 20–25 stimuli were averaged together, and the a- and b-wave responses were acquired from the averaged ERG waveform. The ERGs were analyzed with the Espion V6 software (Diagnosys LLC).

**scRNA-seq analysis**. Mice were euthanized, and eyes were enucleated for retinal tissue isolation. Retinal cells were dissociated using the Papain Dissociation System (Worthington Biochemical) following the manufacturer's instructions and diluted to a final concentration of 1000 cells/μl. In each experimental group, four retinas were used for the mouse scRNA-seq. For each group, freshly dissociated cells (~16,500) were loaded into a 10× Genomics Chromium Single Cell system using v2 chemistry following the manufacturer's instruction. Libraries were pooled and sequenced on Illumina NovaSeq6000 with ~500 million reads per library. Sequencing results were processed through the Cell Ranger 5.0.1 pipeline (10× Genomics) with default parameters. Seurat version 3.1 (90) was used to perform downstream analysis following the standard pipeline using cells with >200 genes and 1000 UMI counts, resulting in 4,240 WT mouse cells, 7482 untreated *rd12* cells, and 5174 treated *rd12* cells. Samples were aggregated, and cell clusters were annotated based on previous literature. UMAP dimension reduction was performed on the top principal components learned from high variance genes. Gene expression of each cell cluster was calculated using the average expression function of Seurat. Gene differential expressions of each cell type among different groups were performed using *FindMarkers* function with Wilcoxon test in Seurat.

**Primary visual cortex (V1) electrophysiology**. Mice were initially anesthetized with 2% isoflurane in a mixture of $N_2O/O_2$ (70%/30%) and then placed into a stereotaxic apparatus. A small, custom-made plastic chamber was glued to the exposed skull. One day after recovery, re-anesthetized animals were placed in a custom-made hammock, maintained under isoflurane anesthesia (1-2% in a mixture of $N_2O/O_2$) and four individual tungsten electrodes were inserted into a small craniotomy above the visual cortex of the right hemisphere. Once electrodes were inserted, the chamber was filled with sterile agar. During recording sessions, animals were sedated with chlorprothixene hydrochloride (1 mg/kg; IM) and kept under light isoflurane anesthesia (0.2–0.4%). EEG and EKG were monitored throughout the experiments and body temperature was maintained with a heating pad (Harvard Apparatus).

Data were acquired using a 32-channel Scout recording system (Ripple). The local field potential (LFP) from multiple locations was band-pass filtered from 0.1 to 250 Hz and stored together with spiking data on a computer with a 1 kHz sampling rate. The LFP signal was cut according to stimulus time stamps and averaged across trials for each recording location to calculate VEPs. The spike signal was band-pass filtered from 500 Hz to 7 kHz and stored in a computer hard drive at 30 kHz sampling frequency. Spikes were sorted online in Trellis (Ripple) while performing visual stimulation. Visual stimuli were generated in Matlab (Mathworks) using Psychophysics Toolbox and displayed on a gamma-corrected LCD monitor (55 inches, 60 Hz; 1920 × 1080 pixels; 52 cd/m² mean luminance). Stimulus onset times were corrected for LCD monitor delay using a photodiode and microcontroller (in-house design).

For recordings of visually evoked responses, cells were first tested with 300 repetitions of a 500 msec bright flash stimulus (105 cd/m²). The background activity was calculated as average activity from 500 msec before stimulus onset for each repetition.

**Analysis of V1 electrophysiology**. The response amplitude of LFP was calculated as a difference between the peak of the positive and negative components in the VEP wave. The response latency was defined as the time point where maximum response occurred. The maximum of the response was defined as the maximum of either the negative or positive peaks. The single-unit responses to the flash stimulus were compared as the maximum response relative to stimulus onset. Average differences between animal groups were considered statistically significant at $P \leq 0.05$ for two-tailed Mann–Whitney $U$-tests. Mean values given in the results include error bars for the standard error of the mean (SEM). All offline data analysis and statistics were performed in Matlab (Mathworks, USA).

**Statistics and reproducibility**. The statistical tests used for each experiment are stated in the corresponding figure captions. Statistical analysis was performed using the GraphPad Prism 9 and Microsoft Excel 2016 software. In vitro experiments were repeated independently at least three times with similar results, as shown in Supplementary Fig. 3a. The experiments in Figs. 1c, d, 5e, and 6e and Supplementary Fig. 5b, c were performed in at least three biologically independent animals and similar results were observed.

**Reporting summary**. Further information on research design is available in the Nature Research Reporting Summary linked to this article.

## Data availability

The main data supporting the results of this study are available within the paper and in Supplementary Information. The deep-sequencing data generated in this study have been deposited in the Sequence Read Archive under accession number PRJNA739996. Source data are provided with this paper.

## Code availability

No custom code or mathematical algorithm was used. The paper does not report the original code.

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

## Acknowledgements

We are grateful for unrestricted grants from Research to Prevent Blindness to the Departments of Ophthalmology at UCI. We thank Dr. William Hauswirth (University of Florida) for sharing reagents and our colleagues at UCI Center for Translational Vision Research and Gavin Herbert Eye Institute for helpful comments regarding this study. This work was supported by NIH grant nos. R01EY009339, R24EY027283, and Research to Prevent Blindness Stein Innovation Award to K.P. E.H.C. was supported by NIH grant nos. T32GM007250 and T32GM008803. S.S. was supported by NIH grant nos. F30EY029136, T32GM007250, and T32EY024236. A.T.F. was supported by National Science Centre Project No. 2019/34/E/NZ5/00434 and the International Centre for Translational Eye Research (MAB/2019/12) project that is carried out within the International Research Agendas program of the Foundation for Polish Science co-financed by the European Union under the European Regional Development Fund. G.A.N. was supported by Helen Hay Whitney Fellowship and HHMI. S.W.D. was supported by T32GM008620. D.R.L. was supported by NIH grant nos. UG3AI150551, U01AI142756, R35GM118062, and RM1HG009490.

## Author contributions

E.H.C., S.S., and K.P. conceived of the strategy and designed the research. E.H.C. and S.S. performed in vitro experiments. G.A.N. and S.K. performed off-target analysis. G.A.N., X.D.G., S.B., and A.R. performed virus cloning, production, and sequencing. E.H.C., S.S., A.T.F., H.L., S.W.D., and Z.D. performed in vivo experiments. T.H. performed scRNA-

seq data analysis. S.S., S.B., D.C.L., D.R.L., and K.P. supervised the project. E.H.C., S.S., and K.P. wrote the manuscript with input from all other authors.

## Competing interests

K.P. is the Chief Scientific Officer of Polgenix Inc. D.R.L. is a consultant and equity owner of Beam Therapeutics, Prime Medicine, and Pairwise Plants, companies that use genome editing. The Broad Institute has filed patent applications on base editing. The remaining authors declare no competing interests.
