## [Peer Review File · Nature Communications]

REVIEWER COMMENTS

Reviewer #1 (Remarks to the Author):

In this report, the author test the efficacy of treating IRD with base editor enzyme using rd12 as the model. By first testing various combination of ABE variants and sgRNA in cell line, the author determined the combination of A6 gRNA with NG-ABE gives the best performance in terms of generating wild-type alleles without by stand editing. When applying this combination in the rd12 retina, an average frequency of 27+/- 12% wild-type transcript is detected. Functional assay of the treated retina indicates that the treatment restores about 30% of the cone function. Furthermore, the rescue effect is long-lasting and can be detected one year after treatment. Overall, the data presented is clear and convincing. The conclusion is well supported, and the manuscript is well writing and easy to follow. My specific comments are the following:

1. This report is a follow-up study of a recent report published by the same group where base editing is demonstrated to be sufficient to rescue rd12 mutant. Therefore, data reported here mainly shows that the rescue persists to 1 year after treatment. Furthermore, slightly higher efficiency can be achieved by adjusting the combination of gRNA and ABEs, from 29% previously to 40%. Since the base editor alters the base in its endogenous locus, it is expected that the rescue would persist. Therefore, the new data offered by this report is overall incremental.
2. The author base editing as a therapeutic method for IRD has advantages over gene therapy. One argument is that gene replacement therapy of RPE65 might not last long term. The author showed the long-term rescue of rd12 by base editing is observed. However, can overexpression of RPE65 in rd12 offer long-term rescue? If yes, the data provided by this study can not be used as evidence that base editing will have a long-last effect in treating RPE65 patients.
3. Base editing is a new and potentially very powerful new tool for targeted gene therapy. I think it is particularly useful for treating dominant gain of function mutations where gene replacement therapy falls short. The work would be more exciting and relevant if the author applies based editing to correct dominant alleles such as the Rho P23H allele.
4. The author should discuss challenges base editing is facing, such as each combination will only target one allele, significantly limited its clinical usage to some high-frequency mutations, the large size of Cas9, making it more difficult to deliver; bystander modifications, etc. These challenges need to be addressed before base editing becomes a widely used approach for therapy.

Reviewer #2 (Remarks to the Author):

There is no doubt the age of precision medicine, through direct correction of genetic variants, is upon us. Choi and colleagues report the profiling of base editors to correct a premature stop mutation in Rpe65. Mutations in RPE65 cause a Leber congenital amaurosis, a rare, but devastating blinding disease. The manuscript is clearly laid out and represents a solid, contribution to the field. After screening a number of base editors in vitro, the authors applied the NG-ABE in vivo, using both lentiviral and AAV.

I have the following minor suggestions and questions which may help improve the manuscript:

- 1) It would be good to clearly state the AAV serotype used (both in the methods and results section)
- 2) Could the data for "allele with bystander base editing" also be shown for AAV?
- 3) I think it would be sensible to "tone down" the speculation of potential clinical utility for dual AAV base editing given the paucity of supportive data for this provided in this manuscript.
- 4) I presume a non-integrating lentivirus was used, and as such did the authors find any evidence for integration in vivo? Could this be investigated for using any data in hand?
- 5) Using the scRNAseq data, could differential gene expression analysis be undertaken between the three groups? It would be interesting to know at the transcriptome-wide level how similar the "treated" rods and cones were to the wild type animals versus the untreated cells.

Reviewer #3 (Remarks to the Author):

In a previous landmark paper, the authors have made the proof-of-concept in vivo of the efficacy of DNA cleavage independent gene editing in a mouse model affected by a nonsense mutation of the Rpe65 gene. An adenine base editing (ABE) strategy was developed. In the present study, the authors aimed to improve the method and to investigate whether ABE can efficiently rescue the cone function and survival, knowing that these cells degenerate quickly in this model. Because cones in human are necessary for central vision (reading, good mobility, color and day vision), it is of prime importance to reveal whether this strategy restores a useful cone function. The previous paper focused on rods, which outnumber the cones by around 20 times in mice, because rods are more useful for a mouse environment.

RPE65 is expressed in the retinal pigment epithelial (RPE) cells and is crucial for the chromophore synthesis which will be transported to the photoreceptors cells to bind opsins and thus to render the cells photosensitive. In consequence, the cells to targets for gene editing are RPE cells.

The first approach was to improve ABE. The authors used three other evolved ABE variants to enlarge the possibility to target non-conventional PAM sequences (NG-ABE, xABE, and NRRH-ABE). NG-ABE was found the most efficient in a cell line expressing the mutant Rpe65 gene, with a correction of around 24% and 4.6% of bystander effect. This approach revealed the importance of screening multiple ABE variants and sgRNAs for each target sequence, as a single base.

Then, in vivo experiments were performed using a lentiviral vector, which is able to contain the mutant Cas9 and the desired gRNA. Based on the observation that S-cones are already affected at 3 weeks of ages (reduced number and S-opsin mislocalization), but are still well present, the researchers decided to administer the treatment in mice at 3 weeks of age, when the retina is fully developed. For the first series of experiments, a lentiviral vector coding for the NG-ABE was subretinally injected and the editing

efficacy evaluated 3 weeks later. As observed in vitro, a sequence correction of about 22% as average was observed on isolated RPE. Such isolation procedure provoked cell loss. A 54% of editing efficacy was observed when eyecup cells are taken altogether. This editing leads to an impressive restoration of retina activity of about 70% (attested by ERG). Then, the researchers moved to the AAV-dual vector, because of its favorable safety profile (AAVs are already used in human retina) to produce the large NG-ABE construct containing also the gRNA. An intein strategy was used allowing protein recombination after the translation process. The AAV-injected mice did not show detectable ERG responses until 7 weeks after injection and the editing efficacy reached 2.7%. The authors claim that this strategy may be applicable for clinical application, but to determine the editing for cone function rescue and survival, a lentiviral approach was then tested.

To study only cones, Rd12 mice were crossed with Gnat1^{-/-} rod deficit mice. Retina and visual cortex activities were recorded using different light stimuli spectrum and intensities. After 6 weeks of treatment, stimulation with UV light clearly demonstrated a restoration of S-activity (30% of Gnat1^{-/-}), which is supported by morphological analyses (performed at 2 month post injection) showing a marked cone rescue in the treated group (S- and M-cones). It is regrettable that the density of S- and M-cones in the Gnat1^{-/-} retina was not presented to estimate the degree of rescue and to correlate this amount with the physiological activity observed. Concerning M-cone activity, the interpretation of the data is in part exaggerated because the authors did not take into account the work of Allen et al PONE 2010 showing that triple knock out mice for Gnat1, Cnga3, and Opn4 have an ERG response for stimuli of similar intensities used in this submitted study. In the Allen's paper, the genetic and pharmacological approaches revealed a contribution of low level of GNAT2 in rods. In the present study, the green light used at 544 nm and a bandwidth of 160 nm suggests that rods can also be stimulated (optimal wavelength being 498 nm). A similar comment can be made for VEP recordings using bright light stimuli. In addition, the response pattern is different in the Rd12;Gnat1^{-/-} treated group in comparison to the Gnat1^{-/-} one and this difference is not discussed or explained.

ERG response improvement was maintained at 6 months in the treated group and the rescued number is very similar in comparison to results obtained at 2 months after the treatment showing the long term efficacy of the treatment

The profile of gene expression after treatment was established by single cell RNAseq. Interestingly, the gene editing leads to an increase expression of specific cone genes such as Arr3, S-opsin, Rbp, Grk1 etc. (not M-opsin) and these data are very coherent with cone function restoration and survival. The characterization of some differentially expressed genes coding for cell survival, death and/or cell stress would have been welcomed to reinforce the output of the strategy used.

This work clearly shows the efficacy of adenine based gene editing in an animal model of LCA2 to rescue cone survival and function and is a stimulating work to translate such approach for other mutant genes in the RPE cells and in the retina in general. However, the paper needs to consider that the Gnat1^{-/-} mouse is an imperfect model to study cones and some experiments need to be adapted to avoid rod contribution after light stimuli. We understand that this model is attractive because rods are not degenerating (or very little) and was thus useful to study the long term efficacy of the ABE on cone survival and function, but the data need to be unequivocal.

Major

The authors need to adapt the parameter of light stimuli to take into account that the rod *Gnat1*^{-/-} mice are in part sensitive to light stimuli (cf Allen et al 2010). A VEP experiment with UV light stimuli would be sufficient to demonstrate the functional connections with the visual cortex. If the text is adapted the experiments on M-opsin cone function can be conserved (for supplemental material?).

For the cone density study after the treatment, a comparison with cone density in the *Gnat1*^{-/-} retina will give more information on the degree of the rescue and may help to better envisage the relationship between cone survival and function.

It would be interesting to present the data of gene expression of genes involved in cone cell death and cell stress if the detection threshold is satisfactory for these genes (see for instance the study of Samardzija et al 2021).

Minor

Line 111 I would suggest to replace “rd12 mice display” by “mouse models of RPE65 deficiency display” to be in line with the cited references.

Reviewer #4 (Remarks to the Author):

Response to authors:

In the manuscript, Choi and Suh et al. investigated whether a selected adenine base editor (ABE) system (NG-ABE + sgRNA A6) can protect retinal cone photoreceptors from further degeneration in rd12 mice, which is a transgenic mouse model for Leber congenital amaurosis (LCA) induced by RPE65 mutations. Based on their previous work, the authors tested variants of ABE + sgRNA combinations, presented data showing subretinal delivery of the selected ABE to rd12 mice resulted in about 40% correction of Rpe65 transcripts and confirmed that ABE could promote cone photoreceptor survival and rescue cone-mediated vision. Because existing AAV-mediated gene therapy, which delivers a full-sized functional copy of the RPE65 transgene to LCA2 patients' RPE cells, may show continuation of retinal degeneration and relapse in visual acuity several years after treatment, the current study provides a solid theoretical basis and experimental evidence for the treatment of LCA in a mouse model.

Major points:

The authors should provide direct comparisons between their results, for example from lentivirus and AAV vectors delivery of NG-ABE+sgRNA A6, with those of FDA approved AAV delivery of full-sized copy of RPE65, in terms of cone survival and vision restoration in rd12 mice. The authors reported subretinal injections of lentivirus vector LV-NG-ABE-A6 showed an average correction frequency of $54 \pm 22\%$, accompanied by 36% M-opsin and 30% S-opsin functional recovery, respectively. Both are very promising, but it's not clear to me whether this represents a substantial improvement over the standard

practice in the field or not. This is of particular importance if the authors are aiming at clinical translation.

More discussion/explanations would be good for the results of dual AAV delivery of NG-ABE and sgRNA A6, in which the author found the correction frequency dropped to $2.7 \pm 1.2\%$. Although the correction rate is anticipated to drop for dual AAV approach, it seems a bit too much. What could account for this drastic drop? Any way to improve? AAV-mediated delivery is approved by FDA in human patients, it would be helpful to find out possible causes for this substantial drop.

In addition, the author showed dual AAV delivery approach had a slower rate towards cone function recovery (Fig. S4, 7 weeks measure by ERG). I am curious whether the authors examined the anatomical data of AAV treated retinas to validate the ERG results, even before 7 weeks. It could be as the authors pointed out, delayed ERG means survived cones can only be functionally detected after 7 weeks, there is a possibility that more cones are protected from degeneration by dual AAV delivery, but for some reason they might not be activated by light stimulus at earlier time points. Showing such data would strengthen the authors' conclusion.

Cautions should be taken when talking about base editing restores cone-mediated visual function in adult rd12Gnat1^{-/-} mice. Rd12Gnat1^{-/-} mice lose about 98% of retinal inputs of rods compared with normal wild-type mice. The drastic changes in retinal dynamics would profoundly impact visual circuits along the retina-LGN-cortex pathway, especially during the developmental critical period. This has been demonstrated by many neurophysiological studies, and oscillatory activities were observed in VEP, indicating intracortical inhibition is heavily affected. Thus data from Gnat1^{-/-} mice are not a proper measure for restoration of vision. I suggest changing the word 'restores' to 'improves'.

Minor Points:

The abstract is a bit too long – Is it within the word limit by Nature Communications?

Fig. 2G, consider a different way to visualize data of ABE-treated groups. It's now too busy to read.

Fig. 5D&E, data of Gnat1^{-/-} should be included in Fig. 5D for consistency between figure panels, and it would help the reader to assess the results of ABE editing.

And for the bottom right panel of Fig. 5E, is the data from dorsal or ventral part of the retina?

Fig. 6, Quantification and statistics of the expression level of cone arrestin should be done in order to support the sentences in lines 272 and 273 of the manuscript.

Figure 6E, 'Untreated' was misspelled.

Figure S6

Significance level should be adequately annotated in figure panels.

REVIEWER COMMENTS

Reviewer #1

In this report, the author test the efficacy of treating IRD with base editor enzyme using rd12 as the model. By first testing various combination of ABE variants and sgRNA in cell line, the author determined the combination of A6 gRNA with NG-ABE gives the best performance in terms of generating wild-type alleles without by stand editing. When applying this combination in the rd12 retina, an average frequency of 27+/- 12% wild-type transcript is detected. Functional assay of the treated retina indicates that the treatment restores about 30% of the cone function. Furthermore, the rescue effect is long-lasting and can be detected one year after treatment. Overall, the data presented is clear and convincing. The conclusion is well supported, and the manuscript is well writing and easy to follow.

My specific comments are the following:

1. This report is a follow-up study of a recent report published by the same group where base editing is demonstrated to be sufficient to rescue rd12 mutant. Therefore, data reported here mainly shows that the rescue persists to 1 year after treatment. Furthermore, slightly higher efficiency can be achieved by adjusting the combination of gRNA and ABEs, from 29% previously to 40%. Since the base editor alters the base in its endogenous locus, it is expected that the rescue would persist. Therefore, the new data offered by this report is overall incremental.

RESPONSE: We sincerely appreciate the reviewer's time and constructive feedback and positive evaluation of our work. We have addressed and clarified all questions in the point-by-point responses, and revised our manuscript accordingly.

2. The author base editing as a therapeutic method for IRD has advantages over gene therapy. One argument is that gene replacement therapy of RPE65 might not last long term. The author showed the long-term rescue of rd12 by base editing is observed. However, can overexpression of RPE65 in rd12 offer long-term rescue? If yes, the data provided by this study can not be used as evidence that base editing will have a long-last effect in treating RPE65 patients.

RESPONSE: We truly appreciate the reviewer's insightful comment. The RPE65 gene replacement therapy is a groundbreaking work from two decades of research, which led to successful clinical trials and commercialization in 2017. It is currently used as a treatment for patients worldwide, whereas base editing is recently introduced as another potential therapeutic approach through proof-of-concept studies in a mouse model. Therefore, we cannot compare the two approaches in terms of long-term effects in humans. Additionally, we do not think that base editing is strictly better than gene replacement therapy. There are advantages and disadvantages in each approach. Our goal in this study is to show that 1) base editing was highly effective in protecting cone photoreceptors in the LCA mouse model and 2) base editing can be an alternative modality for treating inherited ocular diseases, as it allows easy manipulation for targeting a specific mutation with new variants, and for targeting gain-of-function and large-sized gene mutations, which could not be achieved with gene replacement therapy.

The reviewer asked whether gene replacement therapy offers long-term rescue. Based on the clinical reports up to date, it is evident that gene replacement therapy improves the visual sensitivity, which is maintained up to 3 years¹⁻⁴. While there is consensus that the gene replacement therapy improves visual sensitivity in the short term, long-term outcomes on the

natural history of retinal degeneration are still equivocal. We summarized the findings from relevant studies in a table (below).

The long-term clinical trial studies in 2013 and 2015 reported subsequent decline in visual sensitivity and continuous progression of photoreceptor degeneration after 3 years^{3,5,6}. The authors reported that the natural rate of photoreceptor loss due to *RPE65* mutations is not modified by the gene therapy when treatments are initiated after the onset of degeneration, and the visual improvements start waning in the long term^{3,5,6}.

However, clinical studies published in 2019 and 2021 report that visual function improvements after gene therapy were maintained at 4 years, with observation ongoing^{7,8}. Given the short history of the RPE65 gene replacement therapy, it is unclear how long the effects will last at this point. Time will give us the answer with the ongoing follow-up.

What is very encouraging about base editing therapy is its remarkable cone protection in a mouse model of LCA, which was not achieved with gene replacement therapy in a mouse model. Previously published studies in LCA mouse models showed that virus-mediated RPE65 overexpression could partially restore cone ERG function and delay cone degeneration only if administered during retinal development (postnatal day 5) and prior to the onset of degeneration, postnatal day 14^{9,10}. Collectively, these studies showed that gene replacement therapy beyond this time point failed to protect cones from degeneration. Of note, the mouse models used in these studies had functional rod response (*Gnat1*-positive), whereas we performed our study after silencing *Gnat1* to eliminate the signal contribution from rods.

Since no gene replacement therapy has been done with the *rd12Gnat1*^{-/-} mouse model, we injected *AAV2-CBA-hRPE65* (a gift from Dr. William Hauswirth) into *rd12Gnat1*^{-/-} mice at postnatal day 21 (time point used for base editing), and measured cone ERG response and cone density following the same timeline. Consistent with previous studies, we found that gene replacement therapy could not restore cone function and S-cone photoreceptors (**Fig. R1**).

In contrast, base editing treatment was able to successfully halt the progression of cone degeneration when it was administered during the phase of rapid degeneration at postnatal day 21. Although the base editing approach has shown a promising cone-protective effect in mice, its predicted outcome in humans is yet unknown. We should note species-specific differences in the outcomes of gene therapy, including pathophysiological mechanisms, vector tropism, rate of retinal degeneration and anatomy⁶. Despite these differences, there are promising aspects to the base editing approach. First, base editing can introduce a permanent genomic edit, thereby eliminating the possibility of diminishing expression that may occur with exogenous transgene expression⁵. Secondly, a base-edited gene can result in a physiological expression pattern under the control of the endogenous promoter and chromatin context, meeting the metabolic demands of the cells. Lastly, base editing eliminates the expression of the disease-causing truncated, dysfunctional protein, reducing cellular stress.

In conclusion, we acknowledge further needs for extensive studies to optimize safer delivery and evaluate side effects and long-term efficacy of *in vivo* base editing therapy in humans. Outcomes of various gene therapy trials are summarized in the following table.

Study	Conclusion about durability of RPE65 gene therapy
Cidecyan et al. (2013) PNAS	11 subjects. Although gene therapy improved vision for at least 3 years, photoreceptor degeneration progressed unabated in patients after 3 years.

Bainbridge et al. (2015) NEJM	12 subjects. Gene replacement therapy improved retinal sensitivity that peaked at 6-12 months, but subsequently declined. Authors mentioned that the amount of RPE65 required to drive the visual cycle and the demand for RPE65 in affected persons were not met to the extent required for a durable, robust effect.
Jacobson et al. (2015) NEJM	3 subjects. Patients were followed up to 4.5 or 6 years following treatment. All three patients showed decline in vision within 3 years.
Maguire et al. (2019) Ophthalmology	40 subjects. Improvements in light sensitivity were generally maintained between 1 year and 4 years after therapy.
Maguire et al. (2021) Ophthalmology	21 subjects. Many patients showed no significant change in visual sensitivity from years 1 to 3, but some patients showed a decline at year 4. Despite the observed decline between years 3 and 4, these patients maintained a clinically meaningful improvement compared with injection baseline.

Figure R1. Evaluation of cone function and survival in *rd12Gnat1*^{-/-} mice following AAV2-CBA-hRPE65 subretinal injection. (A), Three representative photopic ERG waveforms of M-cones (top) and S-cones (bottom), evoked with flashes of green light and UV light, respectively, from *rd12Gnat1*^{-/-} mice injected with AAV2-CBA-hRPE65 (AAV-RPE65). The red ERG tracing represents the ABE-treated eye for comparison. (B), Average photopic b-wave amplitudes from M-cones (left) and S-cones (right) of *rd12Gnat1*^{-/-} mice injected with AAV-RPE65. Data for untreated and LV-ABE-treated mice are included for comparison. n = 8, each group. n.s., not significantly different. (C), Representative retinal flatmounts from untreated (left) and AAV-RPE65-treated (right) *rd12Gnat1*^{-/-} mouse eyes, labeled with M-opsin (green) and S-opsin (purple) antibodies. Scale bar, 1 mm.

3. Base editing is a new and potentially very powerful new tool for targeted gene therapy. I think it is particularly useful for treating dominant gain of function mutations where gene replacement therapy falls short. The work would be more exciting and relevant if the author applies base editing to correct dominant alleles such as the Rho P23H allele.

RESPONSE: We absolutely agree with the reviewer that base editing has a great potential that can be expanded to numerous genetic disorders caused by different mutations, especially autosomal dominant mutations. We incorporated the reviewer's point in our Discussion.

REVISION:

Original text:

"With these further translational studies, we believe base editing provides new hope for curing inherited blindness."

Revised text:

"With these further developments, base editing will provide a new paradigm for the treatment of numerous inherited ocular diseases caused by different modes of inheritance."

4. The author should discuss challenges base editing is facing, such as each combination will only target one allele, significantly limited its clinical usage to some high-frequency mutations, the large size of Cas9, making it more difficult to deliver; bystander modifications, etc. These challenges need to be addressed before base editing becomes a widely used approach for therapy.

RESPONSE: The reviewer has raised a great point that a lot of autosomal recessive retinal diseases are caused by inheritance of two distinct mutant alleles from each parent, rather than two identical mutant alleles. In order to target multiple genomic sites, different sequence-specific sgRNAs with appropriate base editors must be delivered to cells efficiently while avoiding the increasing risk of off-target genomic damage from multiple genomic integration. In fact, this can be achieved by delivering a pre-assembled base editor ribonucleoprotein (a purified base editor protein complexed with sgRNA) instead of multiple large-sized genes. The delivery of ribonucleoprotein using lipid-like particles or viral-like particles has been demonstrated by several studies¹¹⁻¹³. ABE ribonucleoprotein delivered by a viral-like capsid vector achieved high on-target base editing and undetectable guide-independent off-target activities, thanks to its transient genome editing activity¹¹. Moreover, it was possible to multiplex gene editing by simultaneously delivering different ribonucleoproteins¹³.

In fact, in collaboration with the David Liu Lab (co-author), we are testing the delivery of base editors to mouse RPE cells, using viral-like particles called VLPs. VLPs can efficiently package and deliver adenine base editor protein instead of transgene to the target cells¹³⁻²⁰.

Immunoblot analysis of protein extracts from RPE tissue from VLP-NG-ABE-treated *rd12* mice confirmed the restoration of RPE65 protein without the persistence of ABE protein (**Fig. R2**). Below, we included our preliminary data, setting the stage for follow-up studies.

Figure R2. Immunoblot of protein extracts from RPE tissues of wild-type, untreated, LV-NG-ABE-treated and VLP-NG-ABE-treated mice.

REVISION: In response to the reviewer’s valuable comment, we elaborated in the Discussion section on the milestones that we need to accomplish for clinical translation.

“However, additional preclinical testing in non-human primates, which have foveas, would be necessary before this approach can be tested in patients. Furthermore, alternative delivery methods should be investigated to cover a broad area of the RPE tissue in patients and to circumvent constitutive base editor expression, which could induce unwanted DNA/RNA editing and immune reaction in the long term.”

Reviewer #2

There is no doubt the age of precision medicine, through direct correction of genetic variants, is upon us. Choi and colleagues report the profiling of base editors to correct a premature stop mutation in Rpe65. Mutations in RPE65 cause a Leber congenital amaurosis, a rare, but devastating blinding disease. The manuscript is clearly laid out and represents a solid, contribution to the field. After screening a number of base editors in vitro, the authors applied the NG-ABE in vivo, using both lentiviral and AAV.

RESPONSE: We sincerely appreciate the reviewer’s positive evaluation of our work and constructive feedback. We have addressed and clarified all questions in the point-by-point responses, and revised our manuscript accordingly.

I have the following minor suggestions and questions which may help improve the manuscript:

1) It would be good to clearly state the AAV serotype used (both in the methods and results section)

RESPONSE: We agree with Reviewer 2 that it is important to clearly state the AAV serotype used in the experiment. The original manuscript stated the AAV serotype in the Results section:

“Since adeno-associated virus (AAV) is an ideal vector of choice for gene therapy given its low immunogenicity and favorable safety profile, we also tested targeting the *rd12* mutation by packaging NG-ABE and sgRNA-A6 into AAV. Given the limited packaging capacity of AAV, we took advantage of a split base-editor dual-AAV strategy, in which the ABE is divided into amino-terminal and carboxy-terminal halves and packaged as two separate AAV serotype 2 vectors²¹ (Fig. S4A).”

However, we now have modified the title of the subsection in the Results to make it clear.

REVISION:

Original text:

“AAV-mediated delivery of NG-ABE with sgRNA A6 rescues the phenotype at slower rate”

Revised text:

“AAV2-mediated delivery of NG-ABE with sgRNA A6 rescues the phenotype at a slower rate”

2) Could the data for "allele with bystander base editing" also be shown for AAV?

RESPONSE: We agree with the reviewer’s point. As the reviewer suggested, we analyzed the percentage of modified alleles resulting from dual AAV2 delivery of NG-ABE with sgRNA A6. The percentage of precisely edited target allele was the highest among all modified alleles with 1.6%, followed by alleles with both target and bystander editing at 0.9%, which is consistent with the lentivirus experiment (Fig. S4D). We included the data as Fig. S4D and modified the text in the manuscript. We truly appreciate the reviewer’s thoughtful comment.

REVISION:

Fig. S4. Use of dual-AAV vectors for split base editor delivery. (A), Schematic diagram of split intein-AAV vectors. (B), Scotopic ERG a-wave and b-wave amplitudes recorded at three timepoints in AAV-injected *rd12* mice ($n = 5$ eyes). (C), Frequency of A-to-G conversion in the *Rpe65* genomic DNA isolated from AAV-injected and untreated *rd12* mouse eyes. Bottom sequence represents the 20-nucleotide sgRNA-A6 with the targeted mutation highlighted in red. ABE-treated, $n = 5$; Untreated, $n = 3$. Mean \pm SD. (D), Percentage of modified alleles in each sample used in (C).

3) I think it would be sensible to "tone down" the speculation of potential clinical utility for dual AAV base editing given the paucity of supportive data for this provided in this manuscript.

RESPONSE: We agree with the reviewer's point that the potential utility of dual AAV base editing can be toned down. In response to the reviewer's suggestion, we modified the manuscript accordingly.

REVISION:

Original text:

"However, in clinical practice, dual AAV could be a safe approach for base editor delivery because human retinas do not deteriorate as rapidly and may be rescued despite the slower rate of editing."

Revised text:

"Further optimization of the dual-AAV approach or new development of efficient and safe alternative delivery vehicles would pave the way for translating the base editing technique to patients."

4) I presume a non-integrating lentivirus was used, and as such did the authors find any evidence for integration in vivo? Could this be investigated for using any data in hand?

RESPONSE: Our primary goal in this study was to demonstrate that correction of a point mutation in RPE65 with adenine base editor can rescue cone function and provide sustained protection of cones from progressive degeneration, rather than only to develop a therapy for this single mutation. In this study, we did not use a non-integrating lentivirus.

However, we acknowledge that a delivery method resulting in permanent integration of a transgene into the genome poses a significant safety concern and barrier to clinical adoption. As such, we employed engineered viral-like particles (VLPs) that efficiently package and deliver adenine base editor to the target cells without any genetic material. VLP assemblies of viral proteins that can infect cells, but lack viral genetic material, have emerged as potentially promising vehicles for delivering genome-editing agents as RNPs¹³⁻²⁰.

The immunoblot of protein extracts from RPE tissue from VLP-NG-ABE-treated *rd12* mice confirmed the restoration of RPE65 protein without the persistence of ABE protein (**Fig. R2**). Below, we included the preliminary data, setting the stage for follow-up studies.

Figure R2. Immunoblot of protein extracts from RPE tissues of wild-type, untreated, LV-NG-ABE-treated and VLP-NG-ABE-treated mice. This work has been accepted for publication

22

REVISION: In response to the reviewer’s valuable comment, we elaborated in the Discussion section on the milestones that we need to accomplish for clinical translation.

“However, additional preclinical testing in non-human primates, which have foveas, would be necessary before this approach can be tested in patients. Furthermore, alternative delivery methods should be investigated to cover a broad area of the RPE tissue in patients and to circumvent constitutive base editor expression, which could induce unwanted DNA editing and immune reaction in the long term.”

5) Using the scRNAseq data, could differential gene expression analysis be undertaken between the three groups? It would be interesting to know at the transcriptome-wide level how similar the "treated" rods and cones were to the wild type animals versus the untreated cells.

RESPONSE: We appreciate the reviewer’s great suggestion. To measure the correlations among the transcriptomes of treated, untreated and wild-type, we performed the Pearson’s correlation coefficient analysis. The more similar the expression profiles for all transcripts are between two samples, the higher the correlation coefficient will be. In both cones and rods, we found that the correlation coefficient was higher between WT and Treated than between WT and Untreated (**Fig. S8A, B**), suggesting the treatment induced a global transcriptome change towards the wild-type phenotype.

REVISION: We added the figure shown below as Supplementary Figure 8. We also discussed the finding in the Results section of the main text.

Fig. S8. Heatmap of scRNA-seq correlation between each sample by Pearson's correlation coefficient analysis. (A), cones. (B), rods.

Reviewer #3

In a previous landmark paper, the authors have made the proof-of-concept in vivo of the efficacy of DNA cleavage independent gene editing in a mouse model affected by a nonsense mutation of the Rpe65 gene. An adenine base editing (ABE) strategy was developed. In the present study, the authors aimed to improve the method and to investigate whether ABE can efficiently rescue the cone function and survival, knowing that these cells degenerate quickly in this model. Because cones in human are necessary for central vision (reading, good mobility, color and day vision), it is of prime importance to reveal whether this strategy restores a useful cone function. The previous paper focused on rods, which outnumber the cones by around 20 times in mice, because rods are more useful for a mouse environment.

RESPONSE: We sincerely appreciate the reviewer for highlighting our previous work and understanding the importance of cone survival. As the reviewer described, cone photoreceptors are essential for the highest visual acuity necessary for performing many daily tasks. Thus, it is crucial to develop a treatment that prevents degeneration of cone photoreceptors to sustain vision in patients with inherited retinal disease.

RPE65 is expressed in the retinal pigment epithelial (RPE) cells and is crucial for the chromophore synthesis which will be transported to the photoreceptors cells to bind opsins and thus to render the cells photosensitive. In consequence, the cells to targets for gene editing are RPE cells.

RESPONSE: As the reviewer mentioned, the target cells for ABE-mediated genome editing are the RPE cells because they exclusively express RPE65.

The first approach was to improve ABE. The authors used three other evolved ABE variants to enlarge the possibility to target non-conventional PAM sequences (NG-ABE, xABE, and NRRH-ABE). NG-ABE was found the most efficient in a cell line expressing the mutant Rpe65 gene, with a correction of around 24% and 4.6% of bystander effect. This approach revealed the importance of screening multiple ABE variants and sgRNAs for each target sequence, as a single base.

RESPONSE: As the reviewer described, screening of multiple ABE variants and sgRNAs for a given target sequence is important because this will provide maximized effectiveness of the treatment.

Then, *in vivo* experiments were performed using a lentiviral vector, which is able to contain the mutant Cas9 and the desired gRNA. Based on the observation that S-cones are already affected at 3 weeks of ages (reduced number and S-opsin mislocalization), but are still well present, the researchers decided to administer the treatment in mice at 3 weeks of age, when the retina is fully developed. For the first series of experiments, a lentiviral vector coding for the NG-ABE was subretinally injected and the editing efficacy evaluated 3 weeks later. As observed *in vitro*, a sequence correction of about 22% as average was observed on isolated RPE. Such isolation procedure provoked cell loss. A 54% of editing efficacy was observed when eyecup cells are taken altogether. This editing leads to an impressive restoration of retina activity of about 70% (attested by ERG). Then, the researchers moved to the AAV-dual vector, because of its favorable safety profile (AAVs are already used in human retina) to produce the large NG-ABE construct containing also the gRNA. An intein strategy was used allowing protein recombination after the translation process. The AAV-injected mice did not show detectable ERG responses until 7 weeks after injection and the editing efficacy reached 2.7%. The authors claim that this strategy may be applicable for clinical application, but to determine the editing for cone function rescue and survival, a lentiviral approach was then tested.

RESPONSE: As the reviewer described, we elected to use the lentivirus approach for this study.

To study only cones, Rd12 mice were crossed with *Gnat1*^{-/-} rod deficit mice. Retina and visual cortex activities were recorded using different light stimuli spectrum and intensities. After 6 weeks of treatment, stimulation with UV light clearly demonstrated a restoration of S-activity (30% of *Gnat1*^{-/-}), which is supported by morphological analyses (performed at 2 month post injection) showing a marked cone rescue in the treated group (S- and M-cones). It is regrettable that the density of S- and M-cones in the *Gnat1*^{-/-} retina was not presented to estimate the degree of rescue and to correlate this amount with the physiological activity observed.

RESPONSE: We appreciate the reviewer's valuable suggestion regarding the densities of S- and M-cones in the control *Gnat1*^{-/-} mice for comparison. We further analyzed the densities of S- and M-cones in the age-matched *Gnat1*^{-/-} mice and added these data to the supplementary materials (**Fig. S5C, D**), and modified the text in the manuscript accordingly.

REVISION:

Fig. S6. Comparisons of photopic ERG and cone count to control *Gnat1*^{-/-} mice. (A), Photopic b-wave amplitudes evoked with green light flashes in untreated and treated *rd12Gnat1*^{-/-} mice; and in control *Gnat1*^{-/-} mice. (B), Photopic b-wave amplitudes evoked with UV light flashes in untreated and treated *rd12Gnat1*^{-/-} mice; and in control *Gnat1*^{-/-} mice. *n* = 8, each group. Mean ± SD. **, *P* < 0.01; ***, *P* < 0.001; two-tailed Mann-Whitney U-test for treated *rd12Gnat1*^{-/-} vs. *Gnat1*^{-/-}. (C), Quantification of M-cones in each quadrant at the dorsal and ventral regions of the retina, 1 mm away from the optic nerve. (D), Quantification of S-cones in each quadrant at the dorsal and ventral regions of the retina, 1 mm away from the optic nerve. Five quadrants across the dorsal or ventral retina were analyzed from each eye, with a total of 20 quadrants from 4 eyes per group.

We added the following sentences to the Results:

“Compared to the age-matched *Gnat1*^{-/-}, treated retina protected up to 66% (vs. 30% in untreated) and 26% (vs. 2% in untreated) of M-opsins in the dorsal and ventral region, respectively, and 25% (vs. 5% in untreated) and 17% (vs. 0.2% in untreated) of S-opsins in the dorsal and ventral region (Fig. S6C, D)”

Concerning M-cone activity, the interpretation of the data is in part exaggerated because the authors did not take into account the work of Allen et al PONE 2010 showing that triple knock out mice for *Gnat1*, *Cnga3*, and *Opn4* have an ERG response for stimuli of similar intensities used in this submitted study. In the Allen’s paper, the genetic and pharmacological approaches revealed a contribution of low level of GNAT2 in rods. In the present study, the green light used at 544 nm and a bandwidth of 160 nm suggests that rods can also be stimulated (optimal wavelength being 498 nm). A similar comment can be made for VEP recordings using bright light stimuli.

RESPONSE: We partially agree with the reviewer that this could be a limitation in our data. However, we saw only a tiny residual maximum response of <5 μV to green light stimulation in our pilots with *Rpe65*^{-/-}*Gnat1*^{-/-} mice (Fig. R3). This result is similar to what Kolesnikov *et al.* have previously reported using transretinal *ex vivo* ERG recording with *Rpe65*^{-/-}*Gnat1*^{-/-} mice (Fig. 2B, Kolesnikov...Kefalov 2018, *Scientific Reports*). The *Rpe65*^{-/-}*Gnat1*^{-/-} mouse is phenotypically equivalent to the *rd12Gnat1*^{-/-} mouse.

Fig. R3. Photopic ERG responses to green light stimuli at 300 cd-s/m² in *Rpe65*^{-/-} single mutants (A), and *Rpe65*^{-/-}/*Gnat1*^{-/-} double mutants (B). Note that the residual a-to-b wave response in *Rpe65*^{-/-}/*Gnat1*^{-/-} double mutants is less than 5 μV whereas *Rpe65*^{-/-} single mutants respond maximally at 65-70 μV.

In response to the reviewer's comment, we modified lines 196-198 in the Results section and line 300 in the Discussion with regard to the finding from Allen *et al.* PONE 2010; and to tone down the interpretation about the cone phototransduction. However, since we also saw anatomical rescue of cones, this does not diminish the big picture of the study or its translational impact significantly. Also, the UV light spectrum that we used is quite narrow (**Fig. R4**) and quite far from rhodopsin's absorption spectrum; therefore we are quite confident the UV stimulation activated cone-phototransduction in S cones.

Fig. R4. Emission spectrum of UV LED in the ERG setup.

REVISION: Below is the revised text.

Results, Lines 196-198

Original text:

“...we abolished the rod-mediated photoresponse by crossing the *rd12* mice onto *Gnat1*^{-/-} mice, which lack the rod transducin α -subunit essential for the downstream signal transduction.”

Revised text:

“...we eliminated the majority of the rod-mediated photoresponse by crossing the *rd12* mice onto *Gnat1*^{-/-} mice, which lack the rod transducin α -subunit essential for the downstream signal transduction.”

Discussion, Line 300

Original text:

“...we performed subretinal injections into *rd12Gnat1*^{-/-} mice, which lack the rod-mediated photoresponse, allowing us to measure cone-mediated function alone. Following treatment, we observed by photopic ERG a substantial rescue of M-cone and S-cone function. Furthermore, both M-cones and S-cones were preserved in treated mice up to 6 months of age, and single-cell RNA-seq analysis of treated retinas revealed the restoration of expression of genes associated with cone phototransduction and cone survival. These results support the conclusion that the base editing strategy is able to restore cone function, prolong cone survival, and transform the gene expression signature of early-onset retinal degeneration.”

Revised text:

“...we performed subretinal injections into *rd12Gnat1*^{-/-} mice, which lack the rod-transducin, rendering these mice cone-function dominant. Following treatment, we observed a substantial rescue of ERG activity in response to green and UV monochromatic light stimuli in light-adapted conditions. However, a caveat in using the *Gnat1* knockout strategy pertains to the finding that *Gnat2* (cone transducin) may support some residual rod-function²³, which raises the possibility

that our treatment effect observed on the ERG may be partially mediated by the rods. However, both M- and S-cones were anatomically preserved in treated mice up to 6 months of age, and single-cell RNA-seq analysis of treated retinas revealed the restoration of expression of genes associated with cone phototransduction and cone survival. Collectively, these results support the conclusion that the base editing strategy prolongs cone survival, transforms the gene expression signature of early-onset cone degeneration and supports cone-mediated phototransduction.”

In addition, the response pattern is different in the Rd12Gnat1^{-/-} treated group in comparison to the Gnat1^{-/-} one and this difference is not discussed or explained.

RESPONSE: Thank you for this important question. It is true that the response pattern in the treated *rd12Gnat1^{-/-}* mice is different from that of the *Gnat1^{-/-}* mice. To our knowledge, this was the first time the primary visual cortex recording from *rd12Gnat1^{-/-}* mice has been shown; thus, not much has been reported on the visual synapse in this mouse model. However, we should note that the *rd12Gnat1^{-/-}* mice are very visually disturbed, if not blind, from birth, due to the absence of visual chromophores; and the mouse visual system continues to develop after eye opening. Even if we start the intervention at an early age, the inner retina in *rd12Gnat1^{-/-}* mice may not be correctly formed in postnatal development. The situation is different in *Gnat1^{-/-}* mice, as they have the normal supply of 11-*cis*-retinal from birth.

A good analogy to support this hypothesis is amblyopia, a functional reduction in visual acuity of an eye caused by disuse during visual development. When a child has misaligned eyes, the disuse of the affected eye can lead to inadequate retinal stimulation and abnormal cortical change during a critical phase of visual cortex development²⁴. This can be prevented by patching the normal eye to promote the use of the affected eye. However, studies have shown that treatment outcomes are best when initiated at a very young age while the visual cortex is most malleable²⁵.

REVISION: We appreciate the reviewer’s suggestion to explain the difference, so we added the following discussion:

“However, the VEPs in the treated mice showed attenuated amplitudes and delayed peak times, as compared to control mice (**Fig. 4C, D**). We hypothesize that the absence of visual chromophore in *rd12Gnat1^{-/-}* mice from birth could affect the normal development of visual cortex circuitry during the critical period. Therefore, intervention after the critical period is unable to completely restore normal visual cortical response in *rd12Gnat1^{-/-}* mice. Nevertheless, the activities of single neurons in the primary visual cortex were also improved following the treatment (**Fig. 4E, 4F**).”

ERG response improvement was maintained at 6 months in the treated group and the rescued number is very similar in comparison to results obtained at 2 months after the treatment showing the long term efficacy of the treatment.

RESPONSE: It is correct that we found stable ERG responses when we measured the same treated mice at 2 months and 6 months, which was very encouraging. Although we cannot show the time-dependent change in cone density within the same animal due to the requirement of sacrifice, we could still document the stability of cone survival by showing the average cone density in animals sacrificed at 6 months of age (see table, below).

	Dorsal M opsin	Ventral M opsin	Dorsal S opsin	Ventral S opsin
2 months	426	61	25	123
6 months	375	21	11	74

The profile of gene expression after treatment was established by single cell RNAseq. Interestingly, the gene editing leads to an increase expression of specific cone genes such as Arr3, S-opsin, Rbp, Grk1 etc. (not M-opsin) and these data are very coherent with cone function restoration and survival. The characterization of some differentially expressed genes coding for cell survival, death and/or cell stress would have been welcomed to reinforce the output of the strategy used.

RESPONSE: We appreciate the reviewer's insightful suggestion. To investigate this, we compared genes that were differentially expressed in cones between wildtype, untreated, and treated samples. We would expect genes involved in promoting cone survival to be upregulated in treated samples relative to both wildtype and untreated samples. Conversely, genes involved in cone cell death would be upregulated in untreated cones relative to both wildtype and treated cones. Owing to the relatively small number of cones in the sample, the number of differentially expressed genes identified is far less than seen in previous studies using bulk RNA-Seq of FACS-isolated cones (e.g., Samardzija *et al.* 2021). After three-way comparison and correction for false discovery, a total of 48 differentially expressed genes were identified (FDR<0.05), and are listed in Table S2.

Although this does not include enough genes for meaningful Gene Ontology or signaling pathway-enrichment analysis to be performed, a subset of these genes have been linked to either cell survival, cell death, or stress response. The only candidate neuroprotective gene in this group is *Mt1*, which is strongly and significantly upregulated in treated vs. untreated cones. *Mt2* is also strongly upregulated, but does not reach statistical significance. *Mt1/2* are metal-chelating, antioxidant metallothionein proteins that have been implicated in promoting photoreceptor survival²⁶.

A considerably larger number of genes are upregulated only in untreated retina, and are therefore candidates for promoting photoreceptor cell death. These include the known pro-pyrototic factor *Gsdme*²⁷. The role of other genes in this category are less clear, with several consisting of regulators of cell adhesion (*Itga4*, *Edil3*), intracellular trafficking (*Rebepk*, *Chmp4b*), or isoprenoid biosynthesis (*Pmvk*). Surprisingly, this also includes *Fgf2*, which though known to be induced strongly by photoreceptor injury²⁸ has generally been thought to be neuroprotective.

REVISION: We included a table (labeled Table S2) and supplementary figure (labeled Fig. S9) that show 1) upregulation of genes involved in promoting cone survival in treated vs. untreated mice 2) upregulation of genes promoting photoreceptor cell death in untreated vs. treated mice. We also included these findings in our Results section.

Fig. S9. Dot plot of transcript expression for genes associated with cell survival, death or stress response in cone cells across three groups. *Mt1* and *Mt2* are candidate genes implicated in promoting photoreceptor survival. *Gsdme*, *Fgf2*, *Pmvk*, *Chmp4b*, *Rabepk*, *Edil3* and *Itga4* are candidate genes implicated in promoting photoreceptor cell death or stress response. The shades of red indicate average expression of each gene. The sizes of circles indicate percentage of the cells expressing each gene. All genes except *Mt2* are among top 48 differentially expressed genes identified at FDR < 0.05.

Please refer to the separate supplementary file for the table.

Table S2. List of top differentially expressed genes (FDR <0.05) between treated, untreated and wildtype cone cells.

	p_val	avg_logFC	pct.1	pct.2	p_val_adj	cluster
Fbxw13	6.6E-10	0.259511058	0.23	0.042	2.13E-05	Treated
Opn1sw	1.7E-20	1.410933657	0.671	0.315	5.49E-16	WT
Fbxw211	4.28E-28	0.955148577	0.66	0.152	1.38E-23	Untreated
Kcne2	2.76E-21	0.932118588	0.738	0.391	8.9E-17	WT
Gm17167	7.5E-16	0.547956875	0.309	0.011	2.42E-11	WT
Car8	4.59E-15	0.551496592	0.639	0.242	1.48E-10	Untreated
Ccdc136	7.69E-15	0.921457332	0.682	0.44	2.48E-10	WT
Mpp6	4.86E-14	0.542563326	0.928	0.774	1.57E-09	Untreated
Fgf2	9.23E-13	0.267557435	0.299	0.056	2.98E-08	Untreated
Junb1	1.23E-12	0.68712041	0.753	0.395	3.97E-08	Untreated
Vegfa	1.03E-10	0.594456615	0.499	0.239	3.32E-06	WT
F630040K05Rik	1.31E-10	0.618330991	0.387	0.136	4.21E-06	WT
Gnat2	1.01E-09	0.252432225	0.972	0.962	3.27E-05	WT
Pmvk	2.29E-09	0.455065012	0.412	0.157	7.4E-05	Untreated
Lars2	4.2E-09	0.353012644	0.936	0.88	0.000135631	WT
Gsdme	4.75E-09	0.412614311	0.897	0.758	0.000153241	Untreated
Fbxw21	7.42E-09	0.504126641	0.494	0.195	0.000239707	Treated
Syndig1	7.54E-09	0.463613387	0.306	0.092	0.000243537	WT

Ppia1	1.01E-08	0.436544254	0.845	0.738	0.000325111	Untreated
Mt1	2.65E-08	0.537405204	0.287	0.081	0.000856609	Treated
Rpsa	3.11E-08	0.365810776	0.825	0.742	0.001004874	Untreated
mt-Nd1	3.46E-08	0.293292028	0.959	0.942	0.001115718	Untreated
Nap11	1.04E-07	0.3595312	0.876	0.778	0.0033622	Untreated
Rab6b	1.2E-07	0.343048921	0.577	0.289	0.003864882	Untreated
Fbxw131	1.5E-07	0.25642428	0.196	0.045	0.004848502	Untreated
Plk2	1.7E-07	0.288493995	0.165	0.031	0.005492278	Untreated
Itga4	2.1E-07	0.331385053	0.495	0.233	0.006791926	Untreated
Arr3	2.32E-07	0.270589109	0.992	0.995	0.007500591	WT
Tpt1	2.62E-07	0.343074192	0.876	0.787	0.00844467	Untreated
Cerkl	2.96E-07	0.40040898	0.608	0.339	0.009559277	Untreated
Rpl41	3.47E-07	0.337053794	0.907	0.807	0.011198569	Untreated
Atp2b1	4.06E-07	0.294902969	0.948	0.845	0.013096801	Untreated
Hmgb1	4.37E-07	0.367012642	0.866	0.735	0.014122613	Untreated
Edil3	4.78E-07	0.263457967	0.392	0.157	0.015428256	Untreated
Klhl33	5.04E-07	0.487786714	0.448	0.288	0.016271048	WT
Pdc	5.67E-07	0.260429925	0.911	0.918	0.018319303	WT
Rps16	6.33E-07	0.368018491	0.866	0.7	0.020423119	Untreated
Sgip1	6.53E-07	0.263578401	0.911	0.853	0.02107044	WT
Mfsd4a	6.57E-07	0.330367651	0.443	0.202	0.021207391	Untreated
Rgs9bp	7.16E-07	0.399121346	0.741	0.674	0.02311086	WT
mt-Co1	7.24E-07	0.282103247	0.942	0.957	0.023375713	WT
Rplp1	7.57E-07	0.343610266	0.866	0.794	0.024452998	Untreated
Chgb	7.86E-07	0.352505064	0.856	0.796	0.025368842	Untreated
Rabepk	9.58E-07	0.326719312	0.402	0.17	0.030935991	Untreated
Amer2	1.1E-06	0.413519175	0.845	0.711	0.035534571	Untreated
Ftl1	1.22E-06	0.410101903	0.763	0.601	0.039508596	Untreated
Chmp4b	1.31E-06	0.353417033	0.742	0.52	0.042262555	Untreated
Rabgef1	1.32E-06	0.385182019	0.705	0.56	0.042629917	WT

This work clearly shows the efficacy of adenine based gene editing in an animal model of LCA2 to rescue cone survival and function and is a stimulating work to translate such approach for other mutant genes in the RPE cells and in the retina in general. However, the paper needs to consider that the Gnat1^{-/-} mouse is an imperfect model to study cones and some experiments need to be adapted to avoid rod contribution after light stimuli. We understand that this model is attractive because rods are not degenerating (or very little) and was thus useful to study the long term efficacy of the ABE on cone survival and function, but the data need to be unequivocal.

RESPONSE: We have addressed this comment in the reviewer’s previous comments, and made changes to the manuscript, indicating the potential contribution from residual rod function in *Gnat1*^{-/-} mice.

Major

The authors need to adapt the parameter of light stimuli to take into account that the rod *Gnat1*^{-/-} mice are in part sensitive to light stimuli (cf Allen et al 2010). A VEP experiment with UV light stimuli would be sufficient to demonstrate the functional connections with the visual cortex. If the text is adapted the experiments on M-opsin cone function can be conserved (for supplemental material?).

RESPONSE: As we mentioned in response to the reviewer’s previous comment, we revised our manuscript to indicate that *Gnat1*^{-/-} mice are in part sensitive to light stimuli, with reference to Allen *et al.* PONE 2010. Also, we showed the visual response of the mice to UV light stimuli through ERG, but measuring VEP with UV light was not available with our experimental set-up. In our VEP experiment, visual stimuli are generated in Matlab (Mathworks) using the Psychophysics Toolbox and displayed on a gamma-corrected LCD monitor. Therefore, it does not emit a flash of light, as can be done in the ERG. That’s the reason we used the ERG to measure the S-cone responses to the UV light.

Moreover, the UV light spectrum that we used is quite narrow (peak emission 370 nm, See **Fig. R4**) and quite far from rhodopsin’s absorption spectrum (approximate λ_{max} =498 nm). Therefore, we are quite confident the UV stimulation activated cone-phototransduction in S cones.

Fig. R4. Emission spectrum of UV LED in the ERG setup. Note that this figure is the same as above in the response to another reviewer.

For the cone density study after the treatment, a comparison with cone density in the *Gnat1*^{-/-}

retina will give more information on the degree of the rescue and may help to better envisage the relationship between cone survival and function.

RESPONSE: We appreciate the reviewer's valuable suggestion regarding the densities of S- and M-cones in the control *Gnat1*^{-/-} mice for comparison. We further analyzed the densities of S- and M-cones in the age-matched *Gnat1*^{-/-} mice and added these data to the supplementary materials (**Fig. S6C, D**), and modified the text in the manuscript accordingly.

REVISION:

Fig. S6. Comparisons of photopic ERG and cone count to control *Gnat1*^{-/-} mice. (A), Photopic b-wave amplitudes evoked with green light flashes in untreated and treated *rd12Gnat1*^{-/-} mice; and in *Gnat1*^{-/-} mice. (B), Photopic b-wave amplitudes evoked with UV light flashes in untreated and treated *rd12Gnat1*^{-/-} mice; and in *Gnat1*^{-/-} mice. n = 8, each group. Mean ± SD. **, P < 0.01; ***, P < 0.001; two-tailed Mann-Whitney U-test for treated *rd12Gnat1*^{-/-} mice vs. *Gnat1*^{-/-} mice. (C), Quantification of M-cones in each quadrant at the dorsal and ventral regions of the retina, 1 mm away from the optic nerve. (D), Quantification of S-cones in each quadrant at the dorsal and ventral regions of the retina, 1 mm away from the optic nerve. Five quadrants across the dorsal or ventral retina were analyzed from each eye, with a total of 20 quadrants from 4 eyes per group.

We added the following sentences in the Results:

"Compared to the retinas of age-matched *Gnat1*^{-/-} mice, retinas of the treated *rd12Gnat1*^{-/-} mice showed protection up to 66% (vs. 30% in untreated) and 26% (vs. 2% in untreated) of M-opsins in the dorsal and ventral region, respectively; and 25% (vs. 5% in untreated) and 17% (vs. 0.2% in untreated) of S-opsins in the dorsal and ventral region (**Fig. S6C, D**)."

It would be interesting to present the data of gene expression of genes involved in cone cell death and cell stress if the detection threshold is satisfactory for these genes (see for instance the study of Samardzija et al 2021).

RESPONSE: Please see our detailed response above to the reviewer's comment suggesting a characterization of some differentially expressed genes coding for cell survival, death and/or cell

stress. Owing to the relatively small number of cones in the sample, the number of differentially expressed genes identified is far less than seen in the data from the previous study²⁹ using bulk RNA-Seq of FACS-isolated cones. The sequencing depth and detection threshold is generally insufficient to perform a direct comparison to bulk RNA-seq data from Samardzija *et al.* 2021 and similar studies. An analysis of the Akt, MAPK, and autophagy genes listed as differentially expressed in Figure 6 of Samardzija *et al.* did not yield any clear overlap with differentially expressed genes in this dataset, although the expression levels of all of these genes was very low.

REVISION: We provided detailed revision in our response to the reviewer's previous comment, including **Table S2**, **Fig. S9** and changes to the main text.

Minor

Line 111 I would suggest to replace “rd12 mice display” by “mouse models of RPE65 deficiency display” to be in line with the cited references.

RESPONSE: The reviewer is absolutely right. The cited references include other “mouse models of RPE65 deficiency”, such as *Rpe65*^{-/-} mice. We replaced “rd12 mice” with “mouse models of RPE65 deficiency”.

REVISION: We revised the text accordingly.

Reviewer #4

Response to authors:

In the manuscript, Choi and Suh *et al.* investigated whether a selected adenine base editor (ABE) system (NG-ABE + sgRNA A6) can protect retinal cone photoreceptors from further degeneration in rd12 mice, which is a transgenic mouse model for Leber congenital amaurosis (LCA) induced by RPE65 mutations. Based on their previous work, the authors tested variants of ABE + sgRNA combinations, presented data showing subretinal delivery of the selected ABE to rd12 mice resulted in about 40% correction of Rpe65 transcripts and confirmed that ABE could promote cone photoreceptor survival and rescue cone-mediated vision. Because existing AAV-mediated gene therapy, which delivers a full-sized functional copy of the RPE65 transgene to LCA2 patients' RPE cells, may show continuation of retinal degeneration and relapse in visual acuity several years after treatment, the current study provides a solid theoretical basis and experimental evidence for the treatment of LCA in a mouse model.

RESPONSE: We sincerely appreciate the reviewer's positive evaluation and constructive criticism of our work. We have addressed and clarified all comments in the point-by-point responses, and revised our manuscript accordingly.

Major points:

The authors should provide direct comparisons between their results, for example from lentivirus and AAV vectors delivery of NG-ABE+sgRNA A6, with those of FDA approved AAV delivery of full-sized copy of RPE65, in terms of cone survival and vision restoration in rd12 mice. The authors reported subretinal injections of lentivirus vector LV-NG-ABE-A6 showed an average correction frequency of 54 ± 22%, accompanied by 36% M-opsin and 30% S-opsin functional recovery, respectively. Both are very promising, but it's not clear to me whether this represents a substantial improvement over the standard practice in the field or not. This is of

particular importance if the authors are aiming at clinical translation.

RESPONSE: The reviewer has raised a very important and insightful point. The RPE65 gene replacement therapy is a groundbreaking work from two decades of research, which led to successful clinical trials and commercialization in 2017. It is currently used as a treatment for patients worldwide, whereas base editing is recently introduced as another potential therapeutic approach through proof-of-concept studies in a mouse model. Therefore, we cannot compare the two approaches in terms of long-term effects in humans. Additionally, we do not think that base editing is “better” than gene replacement therapy. There are advantages and disadvantages in each approach. Our goal in this study is to show that 1) base editing was highly effective in protecting cone photoreceptors in the LCA mouse model and 2) base editing can be an alternative modality for treating inherited ocular diseases, as it allows easy manipulation for targeting a specific mutation with new variants, and for targeting gain-of-function and large-sized gene mutations, which could not be achieved with gene replacement therapy.

Based on the clinical reports up to date, it is evident that gene replacement therapy improves the visual sensitivity, which is maintained up to 3 years¹⁻⁴. While there is consensus that the gene replacement therapy improves visual sensitivity in the short term, long-term outcomes on the natural history of retinal degeneration are still equivocal.

The long-term clinical trial studies in 2013 and 2015 reported subsequent decline in visual sensitivity and continuous progression of photoreceptor degeneration after 3 years^{3,5,6}. The authors reported that the natural rate of photoreceptor loss due to *RPE65* mutations is not modified by the gene therapy when treatments are initiated after the onset of degeneration, and the visual improvements start waning in the long term^{3,5,6}.

However, clinical studies published in 2019 and 2021 report that visual function improvements after gene therapy were maintained at 4 years, with observation ongoing^{7,8}. Given the short history of the RPE65 gene replacement therapy, it is unclear how long the effects will last at this point. Time will give us the answer with the ongoing follow-up.

What is very encouraging about base editing therapy is its remarkable cone protection in a mouse model of LCA, which was not achieved with gene replacement therapy in a mouse model. Previously published studies in LCA mouse models showed that virus-mediated RPE65 overexpression could partially restore cone ERG function and delay cone degeneration only if administered during retinal development (postnatal day 5) and prior to the onset of degeneration, postnatal day 14^{9,10}. Collectively, these studies showed that gene replacement therapy beyond this time point failed to protect cones from degeneration. Of note, the mouse models used in these studies had functional rod response (*Gnat1*-positive), whereas we performed our study after silencing *Gnat1* to minimize the signal contribution from rods.

Since no gene replacement therapy has been done with the *rd12Gnat1*^{-/-} mouse model, we injected *AAV2-CBA-hRPE65* (a gift from Dr. William Hauswirth) into *rd12Gnat1*^{-/-} mice at postnatal day 21 (time point used for base editing), and measured cone ERG response and cone density following the same timeline. Consistent with previous studies, we found that gene replacement therapy could not restore cone function and S-cone photoreceptors (**Fig. R1**).

In contrast, base editing treatment was able to successfully halt the progression of cone degeneration when it was administered during the phase of rapid degeneration at postnatal day 21. Although the base editing approach has shown a promising cone-protective effect in mice,

its predicted outcome in humans is yet unknown. We should note species-specific differences in the outcomes of gene therapy, including pathophysiological mechanisms, vector tropism, rate of retinal degeneration and anatomy⁶. Despite these differences, there are promising aspects to the base editing approach. First, base editing can introduce a permanent genomic edit, thereby eliminating the possibility of diminishing expression that may occur with exogenous transgene expression⁵. Secondly, a base-edited gene can result in a physiological expression pattern under the control of the endogenous promoter and chromatin context, meeting the metabolic demands of the cells. Lastly, base editing eliminates the expression of the disease-causing truncated, dysfunctional protein, reducing cellular stress.

In conclusion, we acknowledge further needs for extensive studies to optimize safer delivery and evaluate side effects and long-term efficacy of *in vivo* base editing therapy in humans. Recently, in collaboration with the laboratory group of David Liu (co-author), we have made progress in developing alternative strategies for the *in vivo* delivery of ABE, as the LV-mediated delivery method resulting in permanent integration of a transgene into the genome poses a significant safety concern and barrier to clinical adoption. As such, we employed engineered viral-like particles (VLPs) that efficiently package and deliver ABE protein to the target cells without any genetic material. We present our preliminary data below as **Fig. R2**.

Figure R1. Evaluation of cone function and survival in *rd12Gnat1*^{-/-} mice following AAV2-CBA-hRPE65 subretinal injection. (A), Three representative photopic ERG waveforms of M-cones (top) and S-cones (bottom) evoked with green light and UV light flashes, respectively, from *rd12Gnat1*^{-/-} mice injected with AAV2-CBA-hRPE65 (AAV-RPE65). The red ERG tracing represents the ABE-treated eye for comparison. (B), Average photopic b-wave amplitudes from M-cones (left) and S-cones (right) of *rd12Gnat1*^{-/-} mice injected with AAV-RPE65. Untreated and LV-ABE-treated are included for comparison. n = 8, each group. n.s., not significantly different. (C), Representative retinal flatmounts from untreated (left) and AAV-RPE65-treated (right) *rd12Gnat1*^{-/-} mouse eyes, labeled with M-opsin (green) and S-opsin (purple) antibodies. Scale bar, 1 mm.

Figure R2. Immunoblot of protein extracts from RPE tissues of wild-type, untreated, LV-NG-ABE-treated and VLP-NG-ABE-treated mice.

REVISION: We performed the experiment to satisfy the reviewer's questions. We did not include the above figures in the revised manuscript, as the data are outside the main focus of the present study.

More discussion/explanations would be good for the results of dual AAV delivery of NG-ABE and sgRNA A6, in which the author found the correction frequency dropped to $2.7 \pm 1.2\%$. Although the correction rate is anticipated to drop for dual AAV approach, it seems a bit too much. What could account for this drastic drop? Any way to improve? AAV-mediated delivery is approved by FDA in human patients, it would be helpful to find out possible causes for this substantial drop.

RESPONSE: That is a great question. There are many factors that could contribute to a low correction efficiency with the dual AAV approach, including 1) different viral capsids, 2) requirement for entry of both AAVs into the same cell, 3) requirement for a trans-splicing event to reconstitute C-terminal and N-terminal units, and 4) inevitable dilution of viral titer to deliver two types of AAVs. As seen in **Fig. R5** below, two AAV genomes each containing one half of the expression cassette, should express and reconstitute the full-length transgene expression cassette, which requires trans-splicing technology. The rate of this reconstitution can be significantly influenced by many factors, including the length of the transgene, location of exogenous short recombinogenic sequences, and the biochemistry of intermolecular recombination.

Fig. R5. Split AAV-based strategies to restore large gene expression. Figure derived from Trapani *et al.* Gene Therapy 2021³⁰.

We performed an *in vitro* experiment to compare the relative efficiencies of the dual AAV approach and single lentiviral approach in *rd12* cells. We transiently transfected the expression plasmids, which were used for producing dual AAVs and lentivirus, into *rd12* cells and collected the protein lysates at 48 hr post-transfection. When we first probed for RPE65 expression, RPE65 was rescued from a transfection of a single lentivirus expression plasmid, encoding NG-ABE and sgRNA-A6, but not from two versions of split AAV plasmids (**Fig. R6**). After stripping the membrane, we re-probed with anti-Cas9 and anti- β actin to evaluate the amount of full-size ABE from each group. We only saw the full-length ABE (200 kDa) from the lentivirus-transfected group (lane 5), whereas the AAV-transfected groups (lanes 3 and 4) showed only the C-terminal half (which is detected by antibody). This finding supports the interpretation that reconstitution may be a rate-limiting step that decreases the efficiency of correction. It is possible that the 48 hr duration may not be sufficient for reconstitution in cells, although this can be overcome by *in vivo* transduction. Yet, we can see that dual AAV is not as robust as a single lentivirus in mediating the rescue.

Fig. R6. Transfection of AAV or Lentivirus plasmids into *rd12* cells. Left membrane has been probed with anti-RPE65 antibody. Right membrane is the same membrane which has been stripped and re-probed with anti-Cas9 antibody and anti- β -actin antibody. The table describes the identity of protein lysate in each lane.

REVISION: We performed the experiment to satisfy the reviewer's questions. We did not include the above figures to the manuscript, as the data are outside the main focus of the present study. Instead, we discussed possible causes of low efficiency in dual-AAV approach in the revised manuscript.

In addition, the author showed dual AAV delivery approach had a slower rate towards cone function recovery (Fig. S4, 7 weeks measure by ERG). I am curious whether the authors examined the anatomical data of AAV treated retinas to validate the ERG results, even before 7 weeks. It could be as the authors pointed out, delayed ERG means survived cones can only be functionally detected after 7 weeks, there is a possibility that more cones are protected from degeneration by dual AAV delivery, but for some reason they might not be activated by light stimulus at earlier time points. Showing such data would strengthen the authors' conclusion.

RESPONSE: For clarification, we did not measure cone function in the AAV-injected mice as we did for *rd12Gnat1^{-/-}* mice in **Fig.3D, E**. We injected dual-AAV into the *rd12* mouse model, not the *rd12Gnat1^{-/-}* model, and measured the scotopic ERG response instead of photopic ERG response (which measures cones). The primary purpose of the dual-AAV experiment was to decide whether we should use the dual-AAV vs. the single-lentivirus approach to deliver ABE to investigate its effect on cone function and survival. After we injected either lentivirus or dual-AAV into *rd12* mice and measured the scotopic ERG response from both groups, we found that lentivirus-injected mice showed significantly higher scotopic ERG amplitude (see **Fig. S3B**). Therefore, we decided to use the lentivirus approach for our cone study. That's when we began to measure cone function using the *rd12Gnat1^{-/-}* mouse model and photopic ERG with green and UV stimuli.

Therefore, in our experiment we did not measure the function of cones from the AAV-treated *rd12* mice.

Cautions should be to be taken when talking about base editing restores cone-mediated visual function in adult *rd12Gnat1^{-/-}* mice. *Rd12Gnat1^{-/-}* mice lose about 98% of retinal inputs of rods compared with normal wild-type mice. The drastic changes in retinal dynamics would profoundly impact visual circuits along the retina-LGN-cortex pathway, especially during the developmental critical period. This has been demonstrated by many neurophysiological studies, and oscillatory activities were observed in VEP, indicating intracortical inhibition is heavily affected. Thus data from *Gnat1^{-/-}* mice are not a proper measure for restoration of vision. I suggest changing the word 'restores' to 'improves'.

RESPONSE: This is an important point made by the reviewer, and we have corrected the wording to "improve". We agree that the existing central visual pathway in *Rd12Gnat1^{-/-}* mice may develop differently than a normal mouse with a rod-dominated retina. Therefore, "restore" would not be an accurate description. Nevertheless, how much a cone-dominated retina would affect central circuits in a mouse is unclear and requires further study. As mentioned above, the recovery of cone function is more relevant to the human cone-dominated fovea. When applied to humans this treatment could more appropriately be referred to as restoring visual abilities, as the central pathway would have developed under cone-dominated conditions, at least for the central visual field.

RESPONSE: We made the change from "restores" to "improves".

Minor Points:

The abstract is a bit too long – Is it within the word limit by Nature Communications?

RESPONSE: Thank you for the important point. We revised our abstract to be within the 150-word limit.

REVISION: Please refer to our revised abstract.

Fig. 2G, consider a different way to visualize data of ABE-treated groups. It's now too busy to read.

RESPONSE: We agree with the reviewer's perspective. In fact, we had tried many approaches to best present Fig. 2G in a less cluttered way by using a pie chart or grouped columns. However, we found that a **100% stacked column graph** was the best way to convey the key message of this figure. In this figure, we aim to show the composition of possible *Rpe65* allelic variants in each eye after treatment. While a pie chart would make it easier to visualize, this would not allow us to show a standard deviation among samples. Also, while we can combine two categories into one to make this figure less busy (e.g., combine "target + bystander edit" and "bystander edit" into one big category, "non-precise edit"), we think showing the % of every category is important to convey the base-editing outcome more accurately.

Below, we included a pie chart per reviewer's suggestion (**Fig. R7**). Each pie slice represents the mean without standard deviation. Also, please note that the sum of each group of slices is not 100% due to a small amount of incorrect reads generated during deep sequencing.

Fig. R7. Pie chart showing composition of identified *Rpe65* transcript variants per eye from three groups. Each slice represents the mean. WT, n = 3; Untreated, n = 3; ABE-treated, n = 6.

REVISION: We kept the original column bar graph, as this is the way to display the standard deviations.

Fig. 5D&E, data of *Gnat1*^{-/-} should be included in Fig. 5D for consistency between figure panels, and it would help the reader to assess the results of ABE editing. And for the bottom right panel of Fig. 5E, is the data from dorsal or ventral part of the retina?

RESPONSE: We appreciate the reviewer's great suggestion. We measured the number of cone cells from *Gnat1*^{-/-} mice. We reported this finding separately in Fig. S5D instead of putting it next to the treated and untreated groups in Fig. 5D, because the large value for the *Gnat1*^{-/-} mice makes it difficult to compare the cone densities for treated vs. untreated *rd12Gnat1*^{-/-} mice. The explanation for a remarkable difference between *Gnat1*^{-/-} vs. treated *rd12Gnat1*^{-/-} mice is attributed to early occurrence of cone death in the *rd12Gnat1*^{-/-} mice, even before the treatment. As our Fig. 3B shows, *rd12Gnat1*^{-/-} mice undergo significant cone cell death by 3 weeks of age,

so there are decreased cone cells to begin with. We also discussed the cone density of the *Gnat1*^{-/-} mice in the text.

Thank you for catching the missing details in Fig. 5E. The top panel is dorsal retina to show M-opsins and the bottom panel is ventral retina to show S-opsins. We added these details to our figure legend.

REVISION: We added the location description to the figure legend for Fig. 5E.

Fig. 6, Quantification and statistics of the expression level of cone arrestin should be done in order to support the sentences in lines 272 and 273 of the manuscript.

RESPONSE: The expression level of cone arrestin (*Arr3*) and the p-value for treated vs. untreated are reported in Table S1.

REVISION: Please refer to **Table S1**.

Figure 6E, 'Untreated' was misspelled.

RESPONSE: Thank you for identifying the misspelling. We corrected this in Fig. 6E.

REVISION: Please refer to **Fig. 6E**.

Figure S6
Significance level should be adequately annotated in figure panels.

RESPONSE: Thank you for this comment. We annotated the significance in the figure and added this detail to our figure legends.

REVISION: In **Fig. S6A, B**, we performed an unpaired t-test between *Gnat1*^{-/-} and treated group, and annotated the significance on graph.

References

1. Cideciyan AV, Hauswirth WW, Aleman TS, et al. Human RPE65 Gene Therapy for Leber Congenital Amaurosis: Persistence of Early Visual Improvements and Safety at 1 Year. *Hum Gene Ther.* 2009;20(9):999-1004.
2. Jacobson SG, Cideciyan AV, Ratnakaram R, et al. Gene Therapy for Leber Congenital Amaurosis Caused by RPE65 Mutations: Safety and Efficacy in 15 Children and Adults Followed Up to 3 Years. *Arch Ophthalmol.* 2012;130(1):9-24.
3. Cideciyan AV, Jacobson SG, Beltran WA, et al. Human retinal gene therapy for Leber congenital amaurosis shows advancing retinal degeneration despite enduring visual improvement. *Proc Natl Acad Sci U S A.* 2013;110(6):E517-525.
4. Testa F, Maguire AM, Rossi S, et al. Three-Year Follow-up after Unilateral Subretinal Delivery of Adeno-Associated Virus in Patients with Leber Congenital Amaurosis Type 2. *Ophthalmology.* 2013;120(6):1283-1291.
5. Jacobson SG, Cideciyan AV, Roman AJ, et al. Improvement and decline in vision with gene therapy in childhood blindness. *N Engl J Med.* 2015;372(20):1920-1926.
6. Bainbridge JW, Mehat MS, Sundaram V, et al. Long-term effect of gene therapy on Leber's congenital amaurosis. *N Engl J Med.* 2015;372(20):1887-1897.
7. Maguire AM, Russell S, Chung DC, et al. Durability of Voretigene Neparvovec for Biallelic RPE65-Mediated Inherited Retinal Disease: Phase 3 Results at 3 and 4 Years. *Ophthalmology.* 2021;128(10):1460-1468.
8. Maguire AM, Russell S, Wellman JA, et al. Efficacy, Safety, and Durability of Voretigene Neparvovec-rzyl in RPE65 Mutation-Associated Inherited Retinal Dystrophy: Results of Phase 1 and 3 Trials. *Ophthalmology.* 2019;126(9):1273-1285.
9. Bemelmans AP, Kostic C, Crippa SV, et al. Lentiviral gene transfer of RPE65 rescues survival and function of cones in a mouse model of Leber congenital amaurosis. *PLoS Med.* 2006;3(10):e347.
10. Pang J, Boye SE, Lei B, et al. Self-complementary AAV-mediated gene therapy restores cone function and prevents cone degeneration in two models of Rpe65 deficiency. *Gene Ther.* 2010;17(7):815-826.
11. Lyu P, Lu Z, Cho S-I, et al. Adenine Base Editor Ribonucleoproteins Delivered by Lentivirus-Like Particles Show High On-Target Base Editing and Undetectable RNA Off-Target Activities. *The CRISPR Journal.* 2021;4(1):69-81.
12. Chen G, Abdeen AA, Wang Y, et al. A biodegradable nanocapsule delivers a Cas9 ribonucleoprotein complex for in vivo genome editing. *Nature Nanotechnology.* 2019;14(10):974-980.
13. Hamilton JR, Tsuchida CA, Nguyen DN, et al. Targeted delivery of CRISPR-Cas9 and transgenes enables complex immune cell engineering. *Cell Rep.* 2021;35(9):109207.
14. Choi JG, Dang Y, Abraham S, et al. Lentivirus pre-packed with Cas9 protein for safer gene editing. *Gene Ther.* 2016;23(7):627-633.
15. Lyu P, Javidi-Parsijani P, Atala A, Lu B. Delivering Cas9/sgRNA ribonucleoprotein (RNP) by lentiviral capsid-based bionanoparticles for efficient 'hit-and-run' genome editing. *Nucleic Acids Res.* 2019;47(17):e99.
16. Indikova I, Indik S. Highly efficient 'hit-and-run' genome editing with unconcentrated lentivectors carrying Vpr.Prot.Cas9 protein produced from RRE-containing transcripts. *Nucleic Acids Res.* 2020;48(14):8178-8187.
17. Mangeot PE, Risson V, Fusil F, et al. Genome editing in primary cells and in vivo using viral-derived Nanoblades loaded with Cas9-sgRNA ribonucleoproteins. *Nat Commun.* 2019;10(1):45.

18. Campbell LA, Coke LM, Richie CT, Fortuno LV, Park AY, Harvey BK. Vesicle-Mediated Delivery of CRISPR/Cas9 Ribonucleoprotein Complex for Inactivating the HIV Provirus. *Mol Ther*. 2019;27(1):151-163.
19. Gee P, Lung MSY, Okuzaki Y, et al. Extracellular nanovesicles for packaging of CRISPR-Cas9 protein and sgRNA to induce therapeutic exon skipping. *Nat Commun*. 2020;11(1):1334.
20. Lyu P, Lu Z, Cho SI, et al. Adenine Base Editor Ribonucleoproteins Delivered by Lentivirus-Like Particles Show High On-Target Base Editing and Undetectable RNA Off-Target Activities. *CRISPR J*. 2021;4(1):69-81.
21. Levy JM, Yeh WH, Pendse N, et al. Cytosine and adenine base editing of the brain, liver, retina, heart and skeletal muscle of mice via adeno-associated viruses. *Nat Biomed Eng*. 2020;4(1):97-110.
22. Banskota S, Raguram A, Suh S, et al. Therapeutic in vivo base editing with minimal off-target activity using engineered DNA-free virus-like particles. *Cell*. 2022((in press)).
23. Allen AE, Cameron MA, Brown TM, Vugler AA, Lucas RJ. Visual Responses in Mice Lacking Critical Components of All Known Retinal Phototransduction Cascades. *PLoS One*. 2010;5(11):e15063.
24. Good WV. Cortical visual impairment: new directions. *Optometry and vision science : official publication of the American Academy of Optometry*. 2009;86(6):663-665.
25. Sengpiel F. Plasticity of the visual cortex and treatment of amblyopia. *Curr Biol*. 2014;24(18):R936-r940.
26. Suemori S, Shimazawa M, Kawase K, et al. Metallothionein, an Endogenous Antioxidant, Protects against Retinal Neuron Damage in Mice. *Invest Ophthalmol Vis Sci*. 2006;47(9):3975-3982.
27. McKenzie BA, Fernandes JP, Doan MAL, Schmitt LM, Branton WG, Power C. Activation of the executioner caspases-3 and -7 promotes microglial pyroptosis in models of multiple sclerosis. *J Neuroinflammation*. 2020;17(1):253.
28. Rattner A, Nathans J. The Genomic Response to Retinal Disease and Injury: Evidence for Endothelin Signaling from Photoreceptors to Glia. *The Journal of Neuroscience*. 2005;25(18):4540.
29. Samardzija M, Corna A, Gomez-Sintes R, et al. HDAC inhibition ameliorates cone survival in retinitis pigmentosa mice. *Cell Death Differ*. 2021;28(4):1317-1332.
30. Trapani I, Tornabene P, Auricchio A. Large gene delivery to the retina with AAV vectors: are we there yet? *Gene Ther*. 2021;28(5):220-222.

REVIEWERS' COMMENTS

Reviewer #1 (Remarks to the Author):

The authors had done a great job address the critics raised by the reviewers.

Reviewer #2 (Remarks to the Author):

All changes made by the author appear appropriate and well reasoned. I think this work makes an important contribution to the literature.

Reviewer #3 (Remarks to the Author):

The authors paid attention to all my concerns and responded with satisfactions to all of them. In addition, several sections of the manuscripts were improved and I consider this article well documented with adapted discussion. This work should have a broad interest.

Reviewer #4 (Remarks to the Author):

Response to authors:

The authors have largely addressed my concerns. There is just one thing: in the revised manuscript, the authors explained that their ABE method should not be compared with the FDA-approved gene replacement therapy. I found the explanation not very convincing. The authors should give a better argument on the advantages of their ABE method, for example, in the discussion, otherwise the findings would appear a bit too incremental, as other reviewers have also pointed out.

REVIEWER COMMENTS

Reviewer #1

The authors had done a great job address the critics raised by the reviewers.

RESPONSE: We are very pleased that the reviewer finds our revised manuscript to be satisfactory. We sincerely appreciate the reviewer's time and constructive feedback that played a significant role in improving the quality of our study.

Reviewer #2

All changes made by the author appear appropriate and well-reasoned. I think this work makes an important contribution to the literature.

RESPONSE: We are very pleased that the reviewer finds our changes appropriate and well-reasoned. We sincerely appreciate the reviewer's time and constructive feedback that played a significant role in improving the quality of our study.

Reviewer #3

The authors paid attention to all my concerns and responded with satisfactions to all of them. In addition, several sections of the manuscripts were improved and I consider this article well documented with adapted discussion. This work should have a broad interest.

RESPONSE: We are very pleased that our revised manuscript satisfied all the concerns and showed improvement. We sincerely appreciate the reviewer's time and constructive feedback that played a significant role in improving the quality of our study.

Reviewer #4

The authors have largely addressed my concerns. There is just one thing: in the revised manuscript, the authors explained that their ABE method should not be compared with the FDA-approved gene replacement therapy. I found the explanation not very convincing. The authors should give a better argument on the advantages of their ABE method, for example, in the discussion, otherwise the findings would appear a bit too incremental, as other reviewers have also pointed out.

RESPONSE: We are very pleased that we have largely addressed the reviewer's concerns. In response to the reviewer's one last concern, we included a statement in our letter to the reviewers, but not in our manuscript. Now, in the penultimate paragraph of the Discussion, we describe the unique advantages of the ABE method. It should also be noted that two technologies are at different stages and comparison is difficult to make. The first one, gene augmentation is FDA-approved, whereas the second is in early development to be applied to human conditions.